# Myeloid-derived interleukin-1β drives oncogenic *KRAS*-NF-κB addiction in malignant pleural effusion

Antonia Marazioti[1], Ioannis Lilis[1], Malamati Vreka[1,2], Hara Apostolopoulou[1], Argyro Kalogeropoulou[3], Ioanna Giopanou[1], Georgia A. Giotopoulou[1], Anthi C. Krontira[1], Marianthi Iliopoulou[1], Nikolaos I. Kanellakis[1], Theodora Agalioti[1], Anastasios D. Giannou[1], Celestial Jones-Paris[4], Yoichiro Iwakura [5], Dimitrios Kardamakis[6], Timothy S. Blackwell[4], Stavros Taraviras[3], Magda Spella[1] & Georgios T. Stathopoulos [1,2]

Malignant pleural effusion (MPE) is a frequent metastatic manifestation of human cancers. While we previously identified *KRAS* mutations as molecular culprits of MPE formation, the underlying mechanism remained unknown. Here, we determine that non-canonical IKKα-RelB pathway activation of *KRAS*-mutant tumor cells mediates MPE development and this is fueled by host-provided interleukin IL-1β. Indeed, IKKα is required for the MPE-competence of *KRAS*-mutant tumor cells by activating non-canonical NF-κB signaling. IL-1β fuels addiction of mutant *KRAS* to IKKα resulting in increased CXCL1 secretion that fosters MPE-associated inflammation. Importantly, IL-1β-mediated NF-κB induction in *KRAS*-mutant tumor cells, as well as their resulting MPE-competence, can only be blocked by co-inhibition of both *KRAS* and IKKα, a strategy that overcomes drug resistance to individual treatments. Hence we show that mutant *KRAS* facilitates IKKα-mediated responsiveness of tumor cells to host IL-1β, thereby establishing a host-to-tumor signaling circuit that culminates in inflammatory MPE development and drug resistance.

[1] Department of Physiology, Laboratory for Molecular Respiratory Carcinogenesis, Faculty of Medicine, University of Patras, 26504 Rio, Achaia, Greece. [2] Comprehensive Pneumology Center (CPC) and Institute for Lung Biology and Disease (iLBD), University Hospital, Ludwig-Maximilians University and Helmholtz Zentrum München, Member of the German Center for Lung Research (DZL), 81377 Munich, Bavaria, Germany. [3] Stem Cell Biology Laboratory, Department of Physiology, Faculty of Medicine, University of Patras, 26504 Rio, Achaia, Greece. [4] Division of Allergy, Pulmonary and Critical Care, Department of Internal Medicine, Vanderbilt University School of Medicine, T-1218 MCN, Nashville, TN 37232-2650, USA. [5] Research Institute for Biomedical Sciences, Tokyo University of Science, Tokyo, Chiba 278-0022, Japan. [6] Department of Radiation Oncology and Stereotactic Radiotherapy, Faculty of Medicine, University of Patras, 26504 Rio, Achaia, Greece. Magda Spella and Georgios T. Stathopoulos are co-senior authors. Correspondence and requests for materials should be addressed to A.M. (email: amarazioti@upatras.gr) or to G.T.S. (email: gstathop@upatras.gr)

Malignant pleural effusion (MPE) is one of the most challenging cancer-related disorders. It ranks among the top prevalent metastatic manifestations of tumors of the lungs, breast, pleura, gastrointestinal tract, urogenital tract, and hematopoietic tissues, killing an estimated two million patients worldwide every year and causing 126,825 admissions in U.S. hospitals in 2012 alone[1,2]. The presence of a MPE at diagnosis is an independent negative prognostic factor in patients with lung cancer and mesothelioma[3,4]. In addition, current therapies are non-etiologic and often ineffective, may cause further morbidity and mortality, and have not yielded significant improvements in survival[5,6].

To meet the pressing need for mechanistic insights into the pathobiology of MPE, we developed immunocompetent mouse models of the condition that unveiled inflammatory tumor-to-host signaling networks causing active plasma extravasation into the pleural space[7]. Nuclear factor (NF)-κB activity in tumor cells was pivotal for MPE formation in preclinical models, driving pro-inflammatory gene expression and promoting pleural tumor cell survival[8–10]. However, the mechanism of oncogenic NF-κB activation of MPE-competent pleural tumor cells remained unknown. In parallel, we recently pinned mutant KRAS as a molecular determinant of the propensity of pleural-metastasized tumor cells for MPE formation: mutant KRAS delivered its pro-MPE effects by directly promoting C-C chemokine motif ligand 2 (CCL2) secretion by pleural tumor cells, resulting in pleural accumulation of MPE-fostering myeloid cells[11]. However, a unifying mechanism linking KRAS mutations with oncogenic NF-κB activation and MPE competence of pleural tumor cells was missing.

KRAS mutations have been previously linked to elevated or aberrant NF-κB activity via cell-autonomous and paracrine mechanisms. KRAS-mutant tumors, including lung and pancreatic adenocarcinomas, require active NF-κB signaling[12–14] and NF-κB inhibition blocks KRAS-induced tumor growth[14–16]. In turn, NF-κB activation of KRAS-mutant tumor cells has been associated with enhanced RAS signaling, drug resistance, and stemness[17,18]. Despite significant research efforts, the NF-κB-activating kinases (IκB kinases, IKK) and pathways (canonical, involving IκBα, IKKβ, and RelA/P50, versus non-canonical, comprising IκBβ, IKKα, and RelB/P52) that mediate this oncogenic addiction between mutant KRAS and NF-κB signaling are still elusive and diverse, and different studies indicate that IKKα, IKKβ, IKKγ, IKKε, and/or TANK-binding kinase 1 (TBK1) are key for this[17–24].

Here we use immunocompetent mouse models of MPE to show that mutant KRAS determines the responsiveness of pleural tumor cells to host-delivered interleukin (IL)-1β signals by directly regulating IL-1 receptor 1 (IL1R1) expression. IKKα is further shown to critically mediate IL-1β signaling in KRAS-mutant tumor cells, culminating in marked MPE-promoting effects delivered by C-X-C chemokine motif ligand 1 (CXCL1), and in oncogenic addiction with mutant KRAS evident as drug resistance. Importantly, simultaneous inhibition of IKKα and KRAS is effective in annihilating mutant KRAS-IKKα addiction in MPE.

## Results

### Non-canonical NF-κB signaling of KRAS-mutant cancer cells.
We first evaluated resting-state NF-κB activity of five mouse cancer cell lines with defined KRAS mutations and MPE capabilities in syngeneic C57BL/6 mice[11]: Lewis lung carcinoma (LLC; MPE-competent; $Kras^{G12C}$), MC38 colon adenocarcinoma (MPE-competent; $Kras^{G13R}$), AE17 malignant pleural mesothelioma (MPE-competent; $Kras^{G12C}$), B16F10 skin melanoma, and

PANO2 pancreatic adenocarcinoma (both MPE-incompetent and $Kras^{WT}$) cells. Parallel transient transfection of these cell lines with reporter plasmids encoding Photinus Pyralis LUC under control of either a constitutive (pCAG.LUC) or a NF-κB-dependent (pNF-κB.GFP.LUC; pNGL) promoter[8–11,25] (Fig. 1a) revealed that unstimulated NF-κB activity did not segregate by KRAS mutation status (Fig. 1b). However, when PANO2 cells, a cell line with relatively low NF-κB activity, were transiently transfected with $pKras^{G12C}$, their NF-κB expression levels were elevated (Fig. 1c). Moreover, KRAS mutant (MUT) cells displayed elevated DNA-binding activity of non-canonical NF-κB subunits P52 and RelB by functional NF-κB enzyme-linked immunosorbent assay (ELISA) and enhanced nuclear immunofluorescent localization of RelB compared with $KRAS^{WT}$ cells (Fig. 1d, e). Immunoblotting of cytoplasmic and nuclear extracts revealed that $KRAS^{MUT}$ cells had increased levels of cytoplasmic RelA and IκBα and of nuclear RelB, IκBβ, and IKKα compared with $Kras^{WT}$ cells (Fig. 1f). These results suggest that $KRAS^{MUT}$ cancer cells exhibit non-canonical endogenous NF-κB activity.

### Resistance of KRAS-mutant cancer cells to IKKβ inhibition.
We next examined the effects of small molecule inhibitors of the proteasome (bortezomib[26]), of IKKβ (IMD-0354[27]), or of heat shock protein 90 (HSP90) (17-dimethylaminoethylamino-17-demethoxygeldanamycin (17-DMAG)[28]) that display significant inhibitory activity against IKKβ and/or IKKα (of note, a specific IKKα inhibitor does not exist) on NF-κB reporter activity and cellular proliferation of our murine cancer cell lines (Fig. 1g, h; Supplementary Table 1). Bortezomib, an indirect inhibitor of IKKβ via cytoplasmic accumulation of non-degraded IκBα[16,26], attenuated endogenous NF-κB activity of $Kras^{WT}$ cells but paradoxically activated NF-κB in $KRAS^{MUT}$ cells, at the same time more effectively killing $KRAS^{WT}$ than $KRAS^{MUT}$ cells in vitro. Similarly, IKKβ-selective IMD-0354[27] blocked NF-κB activity and cellular proliferation of unstimulated $KRAS^{WT}$ cells but not of $KRAS^{MUT}$ cells. Interestingly, the HSP90 and dual IKKα/IKKβ inhibitor 17-DMAG[29] was equally effective in limiting NF-κB activity and cellular proliferation of all cell lines irrespective of KRAS mutation status. These results suggest the existence of endogenous resistance of KRAS-mutant cells to IKKβ inhibition, which can be overcome by combined HSP90/IKKα/IKKβ inhibition.

### IL-1-inducible NF-κB activation of KRAS-mutant cancer cells.
We next studied NF-κB activation patterns of our murine cancer cells in response to exogenous stimuli. For this, cells were stably transfected with pNGL, were pretreated with saline or bortezomib (1 μM ~5–10-fold the 50% NF-κB inhibitory concentration obtained from $Kras^{WT}$ cells; Supplementary Table 1), were exposed to 60 different candidate NF-κB-pathway ligands at 1 nM concentration[30], and were longitudinally monitored for NF-κB-dependent LUC activity by bioluminescence imaging of live cells in vitro (Fig. 2a, b; Supplementary Table 2). Incubation with lipopolysaccharide (LPS) and tumor necrosis factor (TNF) resulted in markedly increased NF-κB activity in all cells irrespective of KRAS status, while lymphotoxin β activated NF-κB in all but PANO2 cells, effects that peaked by 4–8 h of incubation and subsided by 16–24 h. Uniquely, IL-1α and IL-β induced NF-κB exclusively in $KRAS^{MUT}$ cells. In addition, bortezomib exaggerated endogenous and inducible NF-κB activation of $KRAS^{MUT}$ cells, in contrast to $KRAS^{WT}$ cells that displayed efficient NF-κB blockade by bortezomib. In line with the above, Il1r1 (encoding IL1R1, cognate to IL-1α/β) expression, but not Tnfrsf1a/Tnfrsf1b (encoding TNF receptors) or Il1a/Il1b expression (that was undetectable in all cell lines), was exclusively restricted to

*KRAS*[MUT] MPE-proficient tumor cells (Fig. 2c, d). We subsequently tested whether inducible NF-κB activation occurs in tumor cells entering the pleural space in vivo, simulating incipient pleural carcinomatosis[4,7]. For this, naive *C57BL/6* mice were pulsed with a million intrapleural p*NGL*-expressing tumor cells

and were serially imaged for NF-κB-dependent bioluminescence. Amazingly, *KRAS*[MUT] MPE-competent cells responded to the pleural environment with markedly escalated NF-κB activity within 4 h after injection, while *KRAS*[WT] MPE-incompetent cells showed diminishing NF-κB signals (Fig. 3a). Interestingly, this

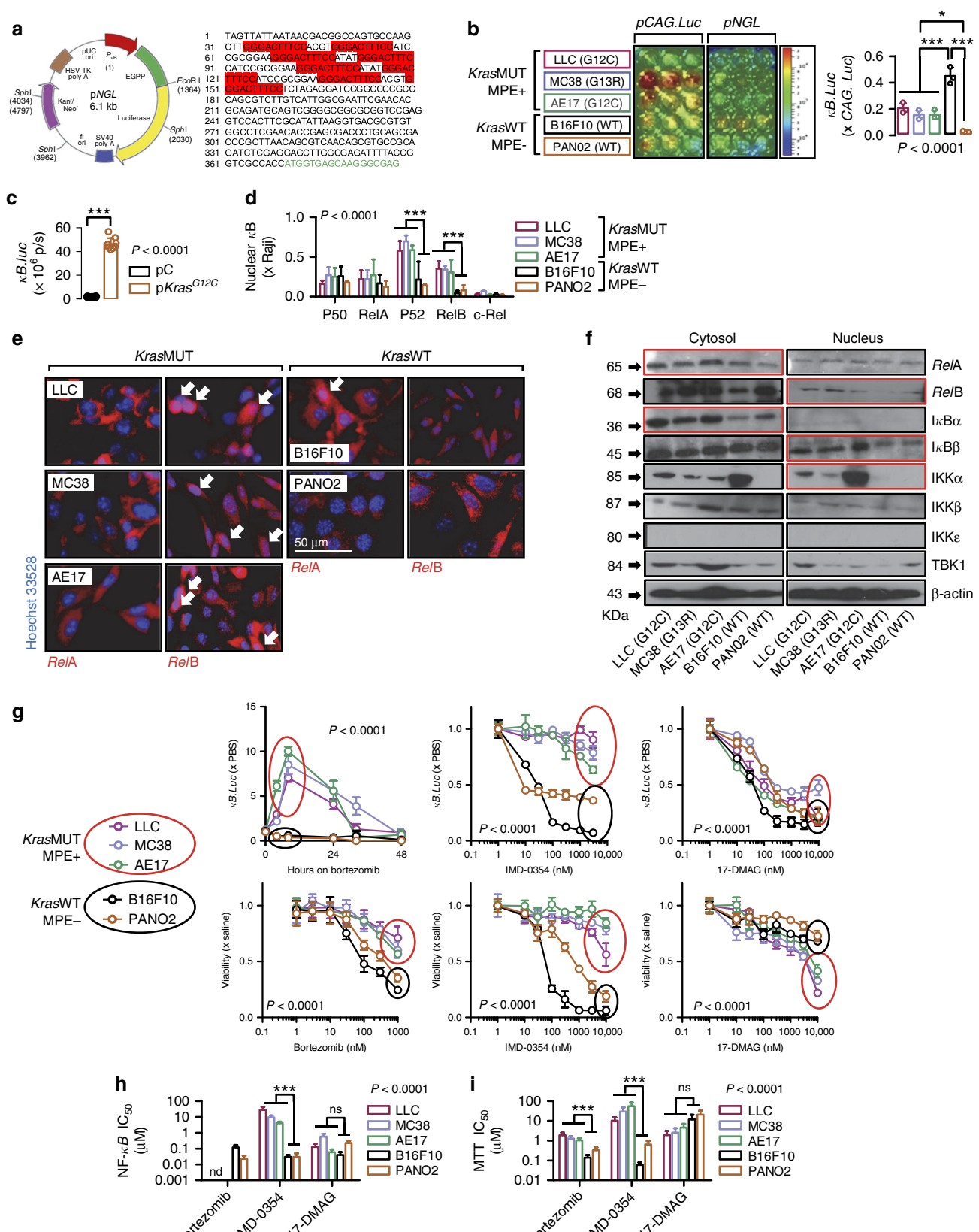

in vivo NF-κB response of $KRAS^{MUT}$ cells was abolished in IL-1β-deficient ($Il1b-/-$[31]), but not in TNF-deficient ($Tnf-/-$[32]), mice (Fig. 3b), indicating that $KRAS^{MUT}$ tumor cells selectively respond to IL-1β of the pleural environment by activating NF-κB.

**Mutant *KRAS* promotes non-canonical NF-κB signaling**. To define the role of mutant *KRAS* in the aberrant NF-κB activation patterns of $KRAS^{MUT}$ tumor cells, including non-canonical endogenous NF-κB activity, resistance to IKKβ inhibition, and IL-1β-inducibility, we undertook short hairpin RNA (shRNA)-mediated *KRAS* silencing (sh*Kras*) and plasmid-mediated over-expression of a mutant dominant-negative form of IκBα (pIκBαDN; inhibits canonical NF-κB signaling) in $KRAS^{MUT}$ cell lines, as well as plasmid-mediated overexpression of mutant *KRAS* (p*Kras*$^{G12C}$) in $KRAS^{WT}$ cell lines[11,33]. Stable pIκBαDN expression in MC38 cells ($Kras^{G13R}$) resulted in decreased *RelA* and sustained *RelB* nuclear-binding activity, while sh*Kras* did not affect *RelA* but abolished *RelB* nuclear-binding activity (Fig. 4a). sh*Kras* also eliminated nuclear *RelB* localization in these cells without affecting *RelA* (Fig. 4b) and abolished nuclear IKKα immunoreactivity of LLC ($Kras^{G12C}$) and MC38 cells (Fig. 4c). sh*Kras* expression reversed the endogenous resistance of MC38 cells to bortezomib and IMD-0354, rendering them as sensitive as $KRAS^{WT}$ cells (Fig. 4d, e). In addition, sh*Kras* annihilated IL-1β-induced NF-κB transcriptional activity of p*NGL*-expressing LLC, MC38, and AE17 ($Kras^{G12C}$) cells (Fig. 4f), and p*Kras*$^{G12C}$ transmitted this phenotype to $Kras^{WT}$ PANO2 cells (Fig. 4g). Importantly, sh*Kras* abrogated the in vivo NF-κB response of pleural-inoculated MC38 cells, which was reinstated in PANO2 cells by stable p*Kras*$^{G12C}$ expression (Fig. 4h, i). In parallel, *KRAS* silencing in $KRAS^{MUT}$ cells significantly decreased, whereas p*Kras*$^{G12C}$ overexpression in $KRAS^{WT}$ cells significantly increased *Il1r1* expression, as well as resting-state and IL-1β-inducible nuclear immunoreactivity for *RelB*, IκBβ, and IKKα (Fig. 4j–l). Collectively, these data indicate that mutant *KRAS* induces non-canonical NF-κB signaling of cancer cells in unstimulated and IL-1β-stimulated conditions.

**IKKα in mutant *KRAS*-dependent MPE**. To define the NF-κB-activating kinase responsible for aberrant NF-κB signaling of $KRAS^{MUT}$ cancer cells, we stably expressed shRNAs specifically targeting IKKα, IKKβ, IKKε, and TBK1 transcripts (*Chuk*, *Ikbkb*, *Ikbke*, and *Tbk1*, respectively) in our p*NGL*-expressing cell lines and validated them (Fig. 5a). In addition, we cloned these murine transcripts into an eukaryotic expression vector and generated stable transfectants of our cell lines. Interestingly, resting-state NF-κB transcriptional activity across $KRAS^{MUT}$ cells was markedly suppressed by sh*Chuk* but not by sh*Ikbkb* or sh*Tbk1*, while

sh*Ikbke* yielded minor NF-κB inhibition in MC38 and AE17 cells. On the contrary, endogenous NF-κB-mediated transcription of B16F10 cells was exclusively silenced by sh*Ikbkb* and, to a lesser extent, sh*Ikbke*, and of PANO2 cells by no shRNA (Fig. 5b). In a reverse approach, overexpression of any kinase resulted in enhanced NF-κB activity in all $KRAS^{MUT}$ cells, of IKKβ only in B16F10 cells, and of no kinase in PANO2 cells (Fig. 5c). In addition to intrinsic, IKKα also mediated IL-1β-inducible NF-κB activity of $KRAS^{MUT}$ tumor cells, since sh*Chuk* but not sh*Ikbkb* abolished IL-1β-induced NF-κB activity across $KRAS^{MUT}$ cell lines (Fig. 5d). In line with the above, sh*Chuk* abolished the immunoreactivity of MC38 cell nuclear extracts for *RelB*, IκBβ, and IKKα, both at resting and IL-1β-stimulated states (Fig. 5e). Taken together, these data suggest that *KRAS*-mutant cancer cells respond to pleural IL-1β via IKKα-mediated non-canonical NF-κB activation. Based on these results and our previous identification of the importance of *KRAS* mutations and NF-κB signaling in MPE development[8–11], we hypothesized that IKKα is required for sustained NF-κB activation and MPE induction by pleural-homed $KRAS^{MUT}$ cancer cells. To test this, we injected IKK-silenced p*NGL*-expressing LLC cells ($Kras^{G12C}$; MPE-competent) into the pleural space of *C57BL/6* mice. Indeed, recipients of IKKα-silenced LLC cells displayed significant reductions in MPE incidence and volume, pleural inflammatory cell influx, and pleural tumor NF-κB activity and prolonged survival. IKKε silencing delivered more modest and equivocal beneficial effects, while IKKβ and TBK1 silencing had no impact (Fig. 6a–c; Supplementary Table 3). These experiments were repeated with IKKα- and IKKβ-silenced MC38 cells ($Kras^{G13R}$; MPE-competent) stably expressing p*NGL*, confirming that IKKα is cardinal for oncogenic NF-κB activation and MPE precipitation by pleural-metastatic $KRAS^{MUT}$ tumor cells (Fig. 6d–f; Supplementary Table 3). However, standalone overexpression of IKKα or IKKβ did not confer MPE competence to $KRAS^{WT}$ PANO2 cells, as opposed to p*Kras*$^{G12C}$ (Fig. 6g–i; Supplementary Table 3), in accord with our previous observations[11]. Collectively, these results suggest that mutant *KRAS*-potentiated IL-1β signaling results in $KRAS^{MUT}$ addiction to IKKα activity, which is required but not sufficient for oncogenic NF-κB activation and MPE formation.

**Myeloid IL-1β fosters mutant *KRAS*-IKKα addiction in MPE**. To study the importance of host-delivered IL-1β in the proposed $KRAS^{MUT}$-IKKα addiction culminating in MPE, we delivered p*NGL*-expressing $KRAS^{MUT}$ LLC and MC38 cells into the pleural space of $Il1b-/-$, $Tnf-/-$, and *WT C57BL/6* mice. Interestingly, $Il1b-/-$ but not $Tnf-/-$ mice displayed decreased MPE incidence, volume, inflammatory cell influx, and oncogenic NF-κB

**Fig. 1** *Kras*-mutant tumor cells exhibit non-canonical endogenous NF-κB activity. Five different *C57BL/6* mouse tumor cell lines with ($Kras^{MUT}$: LLC, MC38, AE17) or without ($Kras^{WT}$: B16F10, PANO2) *Kras* mutations were assessed for activation and inhibition of resting NF-κB activity in vitro. **a** Map of NF-κB reporter plasmid (NF-κB.GFP.Luc; p*NGL*). Partial p*NGL* sequence at origin (1) showing κB-binding motifs (red) and GFP sequence (green). **b** Representative image and data summary ($n = 3$) of area under curve of cumulative bioluminescence emitted by cells transiently transfected with reporter plasmids p*CAG*. *LUC* or p*NGL*. **c** Data summary ($n = 8$) of bioluminescence emitted by PANO2 cells stably expressing p*NGL* reporter plasmid at 48 h after transient transfection with p*C* or p*Kras*$^{G12C}$. **d** Data summary ($n = 5$) of DNA NF-κB motif binding activity of nuclear extracts by NF-κB ELISA relative to nuclear extracts of Raji leukemia cells. **e** Immunofluorescent detection of *RelA* and *RelB* in cells grown on glass slides ($n = 3$) showing increased nuclear localization of *RelB* in $Kras^{MUT}$ cells and of *RelA* in $Kras^{WT}$ cells (arrows). **f** Immunoblots of cytoplasmic and nuclear extracts for NF-κB pathway members and β-actin (representative of $n = 3$ independent experiments). Data presented as mean ± s.d. P, overall probability by one-way (**b**) and two-way (**d**) ANOVA or Student's t-test (**c**). *$P < 0.05$ and ***$P < 0.001$ for the indicated comparisons by Bonferroni post-tests. **g** Data summary ($n = 3$) of p*NGL* reporter activity after 4-h treatment and of cell proliferation by MTT assay after 72-h treatment in response to bortezomib, IMD-0354, or 17-DMAG. Data presented as mean ± s.d. from $n = 3$ replicates/data point. P, probability of no difference between cell lines by extra sum-of-squares F test. **h, i** Data summary of 50% inhibitory concentrations (IC$_{50}$) of NF-κB activity (by p*NGL* reporter activity) and cell proliferation (by MTT; **g**). Data presented as mean ± s.d. from $n = 3$ independent experiments. P, probability of no difference by two-way ANOVA. ns and triple asterisks (***): $P > 0.05$ and $P < 0.001$, respectively, for the indicated comparisons by Bonferroni post-tests. nd not determined

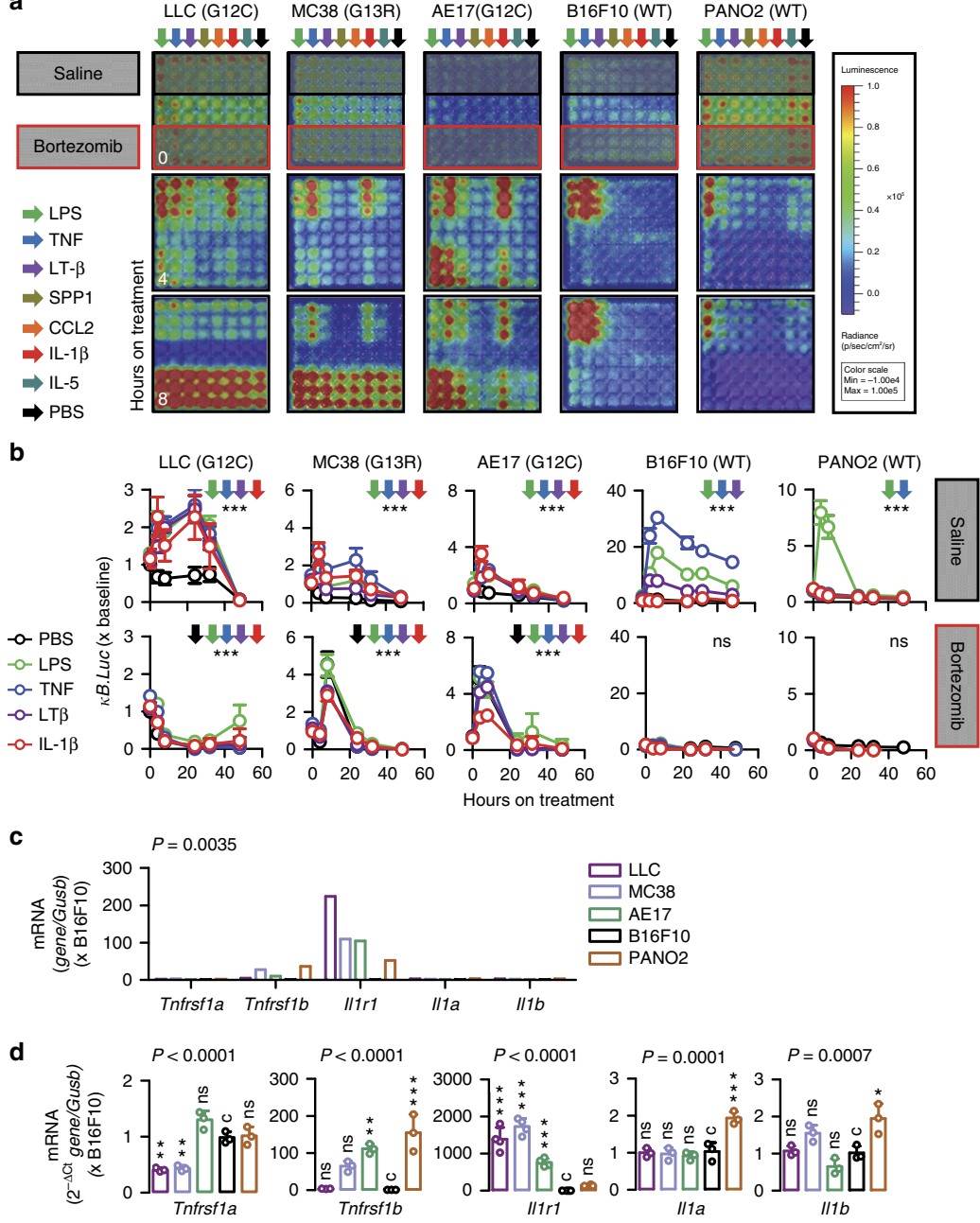

**Fig. 2** *Kras*-mutant tumor cells possess IL-1β-inducible NF-κB activity. Five different *C57BL/6* mouse tumor cell lines with (LLC, MC38, AE17) or without (B16F10, PANO2) *Kras* mutations were assessed for inducible NF-κB activation in response to exogenous stimuli and for the expression of relevant receptors in vitro. **a, b** Representative bioluminescent images (**a**; shown are n = 3 replicates/data-point) and data summary (**b**; mean ± s.d. of n = 3 independent experiments) of cells stably expressing p*NGL* and pretreated with saline or 1 μM bortezomib at different time points after addition of 1 nM of the indicated NF-κB ligands (arrows in **a** and legend in **b**). Note NF-κB inducibility by IL-1β and bortezomib exclusively in *Kras*^MUT cells. ns and *** P > 0.05 and P < 0.001, respectively, for comparison between ligands indicated by colored arrows and PBS at 4 and 8 h on treatment by two-way ANOVA with Bonferroni post-tests. **c, d** *Tnfrsf1a*, *Tnfrsf1b*, *Il1r1*, *Il1a*, and *Il1b* mRNA expression relative to *Gusb* by microarray (**c**) and qPCR (**d**). Shown are mean (**c**) and mean ± s.d. (**d**) of n = 5 independent technical replicates of one biologic sample. P, probability of no difference between cell lines by two-way (**c**) or one-way (**d**) ANOVA. ns, single, double, and triple asterisks (*, **, and ***): P > 0.05, P < 0.05, P < 0.01, and P < 0.001, respectively, for comparison with B16F10 cells (**c**) by Bonferroni post-tests

activation (Fig. 7a–d; Supplementary Table 3). Host-provided IL-1β was of myeloid origin, since bone marrow (BM) transplants[34,35] from *C57BL/6* and *Tnf*−/−, but not *Il1b*−/− donors, to lethally irradiated *Il1b*−/− recipients unable to foster MPE rendered LLC cells MPE proficient (Fig. 7e, f; Supplementary Table 3). To define which myeloid cells provide the bulk of IL-1β to fuel tumor cell NF-κB activity, we isolated BM cells from

*C57BL/6* mice and drove them toward monocyte and neutrophil differentiation by macrophage colony-stimulating factor (M-CSF) and granulocyte-colony-stimulating factor (G-CSF culture, respectively). Both BM-derived monocytes and neutrophils secreted IL-1β upon 24-hour treatment with cell-free LLC supernatants as measured by ELISA, but monocytes secreted ~200 times higher cytokine levels than undifferentiated BM cells

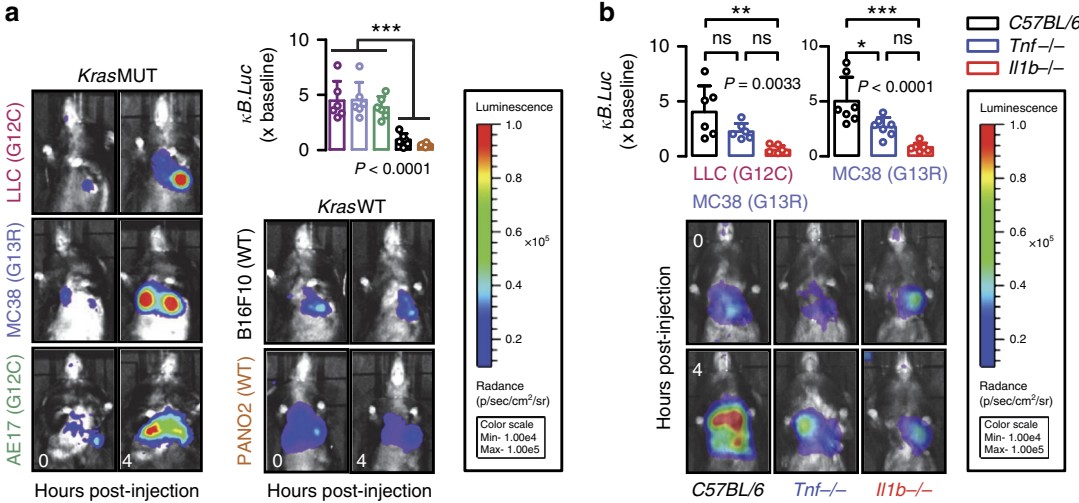

**Fig. 3** Pleural IL-1β activates NF-κB in *Kras*-mutant tumor cells in vivo. Five different *C57BL/6* mouse tumor cell lines with (LLC, MC38, AE17) or without (B16F10, PANO2) *Kras* mutations were assessed for inducible NF-κB activation in response to the pleural environment in vivo. **a** Representative bioluminescent images and data summary (n = 6 mice/cell line) of *C57BL/6* mice at 0 and 4 h after intrapleural injection of a million mouse tumor cells stably expressing p*NGL*. Note the marked induction of the bioluminescent signal emitted specifically by *Kras*[MUT] cells after 4 h. Note also the diminishing signal emitted by *Kras*[WT] cells. **b** Representative bioluminescent images and data summary (LLC: n = 6 mice/genotype; MC38: n = 7 mice/genotype) of *C57BL/6*, *Tnf−/−* and *Il1b−/−* mice at 0 and 4 h after intrapleural injection of LLC or MC38 cells stably expressing p*NGL*. Note the marked induction of the bioluminescent signal in *C57BL/6* mice, the borderline reduction of its inducibility in *Tnf−/−* mice, and the disappearance of signal inducibility in *Ilb−/−* mice. Data are presented as mean ± s.d. *P*, probability of no difference between cell lines or genotypes by one-way ANOVA. ns, single, double, and triple asterisks (\*, \*\*, and \*\*\*): *P* > 0.05, *P* < 0.05, *P* < 0.01, and *P* < 0.001, respectively, for the indicated comparisons by Bonferroni post-tests

and neutrophils (Fig. 7g). These data clearly show that the main source of IL-1β in the pleural space during MPE development likely are recruited myeloid monocyte cells.

**Mutant KRAS-IKKα addiction promotes MPE via CXCL1 secretion**. To identify the MPE effectors and transcriptional signatures of IL-1β/*KRAS*/IKKα-addicted tumor cells, we subjected KRAS-silenced, IKKα-silenced, and IL-1β-challenged LLC and MC38 cells to microarray analyses, seeking for transcripts altered heterodirectionally by silencing/challenge. Thirty transcripts fulfilled these criteria in LLC (including *Ppbp*, encoding pro-platelet basic protein, PPBP, and *Cxcl1*, encoding CXCL1) and 20 in MC38 (including *Cxcl1*) cells, with *Cxcl1* being the only common gene of these two signatures (Fig. 8a, b; Supplementary Tables 4, 5). *Cxcl1* microarray results were validated by quantitative PCR (qPCR) and ELISA (Fig. 8c–e). Furthermore, chromatin immunoprecipitation (ChIP) was performed in LLC cells treated with phosphate-buffered saline (PBS) or IL-1β in order to specify whether and which NF-κB component directly binds to the promoter region of *Cxcl1*. The data indicate that only *Rel*B and IKKα bind to the NF-κB element in the *Cxcl1* promoter and that IL-1β significantly strengthens this binding (Fig. 8f). These findings are consistent with the enhanced transcriptional induction of *Cxcl1*. Moreover, *Cxcl1* and *Ppbp* expression was pivotal for MPE induction by IL-1β/*KRAS*/IKKα-addicted LLC cells, since these were MPE incompetent in both C-X-C chemokine motif receptor 1 (CXCR1) and CXCR2 gene-deficient mice[36,37] that lack the genes encoding CXCL1/PPBP-cognate CXCR1 and CXCR2 receptors[38] (Fig. 8g; Supplementary Table 3). Notably, in MPEs from CXCR1 and CXCR2 gene-deficient mice the predominant cell population was monocytes, whereas in MPEs from CCR2 gene-deficient mice[11] the prevalent cell type was neutrophils. This result was not unexpected since the majority of myeloid cells recruited in the pleural space during MPE development in *C57BL/6* mice consist of both neutrophils and monocytes (Fig. 8h). Of note, the monocyte population is the most prevalent during MPE development.

**Combined targeting of KRAS/IKKα is effective against MPE**. To explore the therapeutic implications of the proposed mechanism, we examined potential synergy of the KRAS inhibitor deltarasin[39] with the IKKβ-specific inhibitor IMD-0354 or the HSP90/IKKα/IKKβ inhibitor 17-DMAG using TNF- or IL-1β-stimulated LLC murine and A549 human lung adenocarcinoma cells expressing p*NGL* (Fig. 9a, b). Interestingly, all inhibitors alone or in combination failed to block TNF-inducible NF-κB activation in both cell lines. In addition, all standalone drugs failed to inhibit IL-1β-inducible NF-κB activation in both cell lines, except from partial effects observed in A549 cells by 17-DMAG. However, deltarasin/17-DMAG but not deltarasin/IMD-0354 combination treatment completely abolished IL-1β-induced NF-κB activation in both cell types to unstimulated levels (Fig. 9a, b), indicating that drugging the *KRAS*/IKKα axis can halt IL-1β responsiveness. To determine the potential efficacy of this approach against MPE, standalone or combined deltarasin, and 17-DMAG treatments (both 15 mg/Kg) were delivered to mice with established pleural tumors. For this, *C57BL/6* mice received pleural LLC cells and treatments commenced after 5 days to allow initial pleural tumor implantation[11]. At day 13 post-tumor cells, standalone deltarasin and 17-DMAG-treated mice had significantly decreased MPE volume compared with saline-treated controls (40% reductions for both groups; *P* < 0.05; one-way analysis of variance (ANOVA) with Bonferroni post-tests). However, combination-treated mice were markedly protected from MPE development (57% incidence) and progression (65% volume reduction; *P* < 0.001; one-way ANOVA with Bonferroni post-tests) (Fig. 9c; Supplementary Table 3). Hence combined targeting of mutant *KRAS* and IKKα is effective in halting oncogenic NF-κB activation and MPE in mice.

**IL-1β-inducible NF-κB activity in human KRAS-mutant cells**. To assess whether our findings are relevant to human cancer, we screened nine human cancer cell lines of known *KRAS* mutation status[40] for *Rel*-binding activity of nuclear extracts. In accord with murine data, *KRAS*[MUT] cells displayed enhanced nuclear *Rel*B

compared with *Rel*A binding (Fig. 10a). In addition, A549 (*KRAS*$^{G12S}$) and NCI-H23 (*KRAS*$^{G12C}$) cells displayed IL-1β-induced NF-κB activation, as opposed to HT-29 and SKMEL2 cells (both *KRAS*$^{WT}$). Importantly, stable p*Kras*$^{G12C}$ expression in SKMEL2 cells rendered them responsive to IL-1β (Fig. 10b, c).

In summary, *KRAS* mutations alter NF-κB signaling in tumor cells. *KRAS*$^{WT}$ cells preferentially utilize intrinsically or exogenously (i.e., by LPS, TNF) stimulated IKKβ-mediated NF-κB signaling, display sensitivity to IKKβ inhibition, poor CXCR1/2 ligand secretion, and MPE incompetence. *KRAS*$^{MUT}$ cells

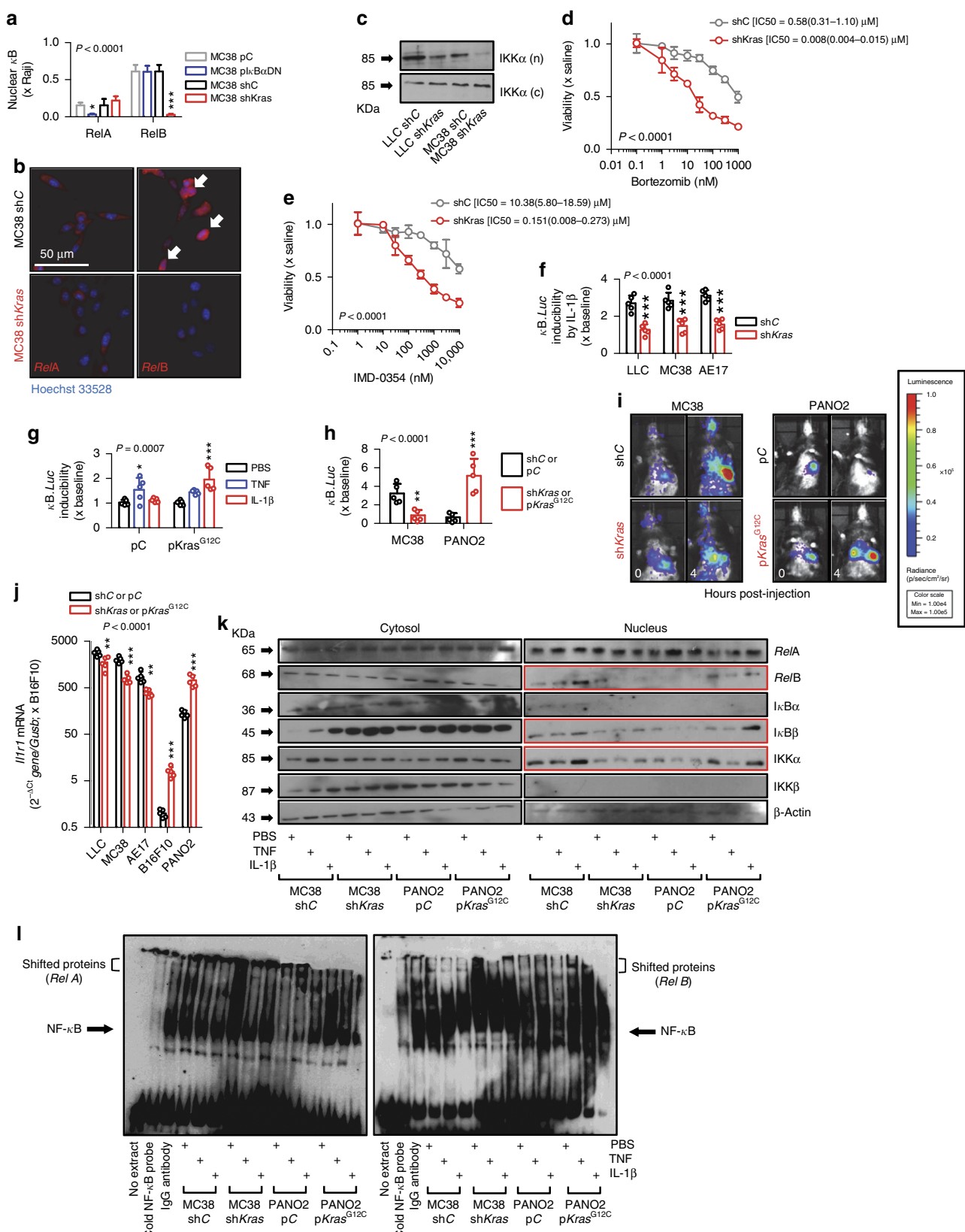

predominantly use IKKα-mediated non-canonical NF-κB signaling at resting state and in response to myeloid IL-1β, display enhanced CXCR1/2 ligand secretion and MPE proficiency, and are addicted to sustained IKKα activity evident as resistance to IKKβ inhibitors.

## Discussion

We provide a novel paradigm of how an oncogene can co-opt the host environment to foster addiction with a perturbed signaling pathway. KRAS-mutant cancer cells are shown to respond to host provided IL-1β in the pleural space by increasing non-canonical IKKα-RelB pathway activity. The co-existence of mutant KRAS and elevated IKKα-mediated non-canonical NF-κB signaling in the cancer cell, relentlessly driven by host IL-1β, leads to two important consequences. First, to enhanced transcription of CXCL1/PPBP chemokines, recruitment of CXCR1+ and CXCR2+ myeloid cells, and frank escalation of inflammatory MPE development. Second, to oncogenic addiction between mutant KRAS and IKKα that culminates in drug resistance. Using immunocompetent mouse models of MPE, we show how IL-1β, mutant KRAS, and IKKα interplay to mediate non-canonical NF-κB activation, resistance to proteasome and IKKβ inhibitors, CXCL1/PPBP secretion, and MPE. Finally, we show that this partnership can be annihilated by combined inhibition of KRAS together with IKKα but not alone.

Although cell-autonomous pro-tumorigenic functions of mutant KRAS are well charted[41–43], mechanisms utilized by the oncogene to co-opt host cells from the tumor microenvironment in order to favor tumor progression have only recently begun to be elucidated. In this regard, mutant KRAS was first shown to promote chemokine secretion by tumor-initiated cells, thereby promoting tumor-associated inflammation[44,45]. Along similar lines, we recently showed that the oncogene is responsible for CCL2 secretion by pleural metastatic cancer cells, fostering inflammatory MPE formation[11]. Our present findings expand the paradigm of how mutant KRAS impacts tumor–host interactions: it renders tumor cells capable of sensing inflammatory IL-1β signals originating from the CCL2-recruited monocytes. The increased Il1r1 expression in these cells could be a result of IL-1β-induced phosphorylation by nuclear IKKa of Ser10 in histone H3 that could be especially important for subsequent modifications in a variety of genes, including Il1r1. Moreover, integrated by IKKα-mediated non-canonical NF-κB activity, IL-1β signaling culminates in enhanced CXCL1/PPBP expression and secretion that function to escalate tumor-associated inflammation required for MPE. Hence, in addition to directly promoting chemokine expression, mutant KRAS is shown here to amplify host-originated inflammatory signals in order to escalate MPE-promoting inflammation.

Mutant KRAS is known to enhance oncogenic NF-κB activity; however, it was mainly linked to IKKβ, IKKε, and TBK1 function[12,14,17,18,21,23,24,43], and only two studies identified IKKα as an accessory to IKKβ in KRAS-mutant lung adenocarcinoma[22] and epidermal growth factor receptor-driven head and neck cancers[19]. Here we show for the first time that KRAS-mutant cancer cells display altered NF-κB utilization in resting and stimulated states, a phenomenon previously identified in pancreatic β cells[46]. Indeed, KRAS-mutant cancer cells displayed non-canonical endogenous NF-κB activity evident by enhanced nuclear localization and/or DNA-binding activity of RelB, IκBβ, and IKKα, which was further inducible by exogenous IL-1β. Importantly, non-canonical NF-κB utilization by KRAS-mutant cancer cells was IKKα driven, involved RelB activation, and was required for MPE. Nuclear IKKα functions have been identified previously, including histone 3 modifications augmenting TNF and receptor activator of NF-κB ligand-induced gene expression and repression of maspin, a metastasis gate-keeper[47–49]. Our work links IKKα function with IL-1β-induced RelB activation and CXCL1/PPBP transcription. Moreover, we provide novel evidence that mutant KRAS is indirectly responsible for non-canonical NF-κB activation, which is IKKa and RelB based, via sensitization of cancer cells to host IL-1β. Finally, IKKα is found to be responsible for MPE, an important metastatic manifestation of various cancers. The findings concur with previous reports of a combined requirement for IKKα and IKKβ for oncogenic NF-κB activation[19,22], as well as with human observations of predominant non-canonical NF-κB activity of tumors with high incidence of KRAS mutations, such as lung adenocarcinoma[50]. However, we demonstrate an isolated requirement for IKKα in KRAS-driven MPE, an important cancer phenotype.

In recent years, inflammation was established as a conditional tumor promoter[51]. IL-1α/β are important components of the tumor microenvironment that stimulate tumor invasiveness and angiogenesis[52]. Myeloid-derived IL-1β is implicated in the resistance to NF-κB inhibitors and IL-1β antagonism yielded beneficial effects in a mouse model of KRAS-mutant pancreatic cancer[53,54]. We found previously that IL-1α/β are present in human and experimental MPE and that MPE-competent adenocarcinomas trigger myeloid cells to secrete IL-1β[35]. Here the mechanism of pleural IL-1β function in MPE promotion is elucidated: CCL2-attracted monocyte-released IL-1β fosters NF-κB activation of MPE-prone KRAS-mutant carcinomas by potentiating non-canonical NF-κB signaling via IKKα. Undoubtedly, IL-1β is not the sole NF-κB ligand expressed in the malignancy-affected pleural space: TNF, a known stimulator of canonical NF-κB signaling, is present in MPE and promotes disease progression[9]. However, TNF likely originates from tumor cells in MPE[9] and non-specifically triggers NF-κB activation in any tumor type irrespective of its KRAS status and MPE competence, suggesting

**Fig. 4** Mutant Kras drives basal and IL-1β-induced non-canonical NF-κB signaling and drug resistance. **a** RelA and RelB binding of nuclear extracts of MC38 cells stably expressing a control plasmid (pC), a mutant dominant-negative form of IκBα (pIκBαDN), control shRNA (shC), or anti-Kras shRNA (shKras) relative to Raji leukemia cells by NF-κB ELISA (n = 3 experiments). **b** Immunofluorescent detection of RelA and RelB in MC38 cells showing increased nuclear RelB (arrows) and its disappearance in cells expressing shKras. **c** IKKα immunoblots of cytoplasmic and nuclear extracts of LLC and MC38 cells expressing shC or shKras. (n = 3 experiments). **d, e** MTT data (n = 3 replicates/data-point) and mean (95% CI) IC$_{50}$ values (n = 3 experiments) of MC38 cells stably expressing shC or shKras treated with bortezomib (**d**) or IMD-0354 (**e**) for 72 h. P, probability of no difference between cell lines by extra sum-of-squares F test. **f, g** Bioluminescent detection of NF-κB activity in Kras$^{MUT}$ (**f**) and Kras$^{WT}$ (**g**) cells stably expressing pNGL and the indicated vectors during 4-h incubation with PBS or 1 nM TNF or IL-1β (n = 3 experiments). **h, i** Data summary (**h**; n = 6 mice/group) and images (**i**) of C57BL/6 mice at 0 and 4 h after intrapleural injection of MC38 or PANO2 cells stably expressing pNGL and the indicated vectors. **j** Il1r1 mRNA expression by qPCR of Kras$^{MUT}$ and Kras$^{WT}$ cells stably expressing the indicated vectors. **k** Immunoblots of protein extracts of MC38 and PANO2 cells stably expressing the indicated vectors for NF-κB members after 4-h incubation with PBS or 1 nM TNF or IL-1β (n = 3 experiments). **l** The above extracts were subjected to EMSA. Super-shift EMSA was performed with the indicated antibodies. IgG antibody served as negative control. Data are presented as mean ± s.d. P, probability of no difference between cell lines by two-way ANOVA. Single, double, and triple asterisks (*, **, and ***): P < 0.05, P < 0.01, and P < 0.001, respectively, for comparison with pC or shC (**a, f, h, j**) or with PBS (**g**) by Bonferroni post-tests

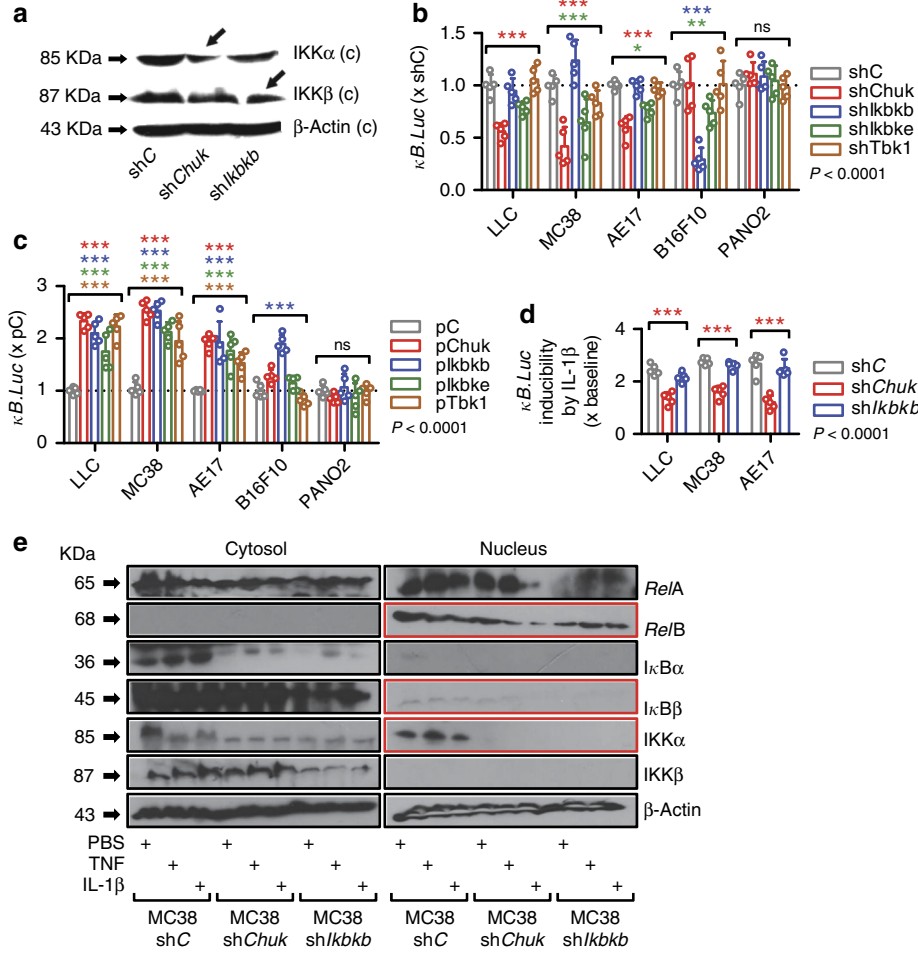

**Fig. 5** IL-1β-induced NF-κB signaling of *KRAS*-mutant cells is IKKα dependent. Five different *C57BL/6* mouse tumor cell lines with (LLC, MC38, AE17) or without (B16F10, PANO2) *Kras* mutations were stably transfected with pNGL NF-κB reporter and any of the following: control shRNA (shC) or shRNA targeting IKKα (sh*Chuk*), IKKβ (sh*Ikbkb*), IKKε (sh*Ikbke*), or TBK1 (sh*Tbk1*) transcripts; control plasmid (pC); or overexpression vectors encodong IKKα (p*Chuk*), IKKβ (p*Ikbkb*), IKKε (p*Ikbke*), or TBK1 (p*Tbk1*) transcripts. **a** Immunoblot of cytoplasmic protein extracts from LLC cells stably expressing shC, sh*Chuk*, or sh*Ikbkb* for IKKα and IKKβ relative to β-actin (representative of $n = 3$ independent experiments). **b** Bioluminescent quantification of NF-κB reporter activity of pNGL cell lines stably expressing shC, sh*Chuk*, sh*Ikbkb*, sh*Ikbke*, and sh*Tbk1*, ($n = 3$ independent experiments). **c** Bioluminescent quantification of NF-κB reporter activity of pNGL cell lines stably expressing pC, p*Chuk*, p*Ikbkb*, p*Ikbke*, and p*Tbk1*, ($n = 3$ independent experiments). **d** Bioluminescent detection of NF-κB reporter activity in LLC, MC38, and AE17 cells (*Kras*^MUT) stably expressing pNGL and shC, sh*Chuk*, or sh*Ikbkb* ($n = 3$ independent experiments) during 4-h incubation with 1 nM IL-1β. Note IL-1β-induced NF-κB activity of shC and sh*Ikbkb* cells that is silenced in sh*Chuk* cells. **e** Immunoblots of cytoplasmic and nuclear protein extracts of MC38 cells stably expressing pNGL and shC, sh*Chuk*, or sh*Ikbkb* after 4-h treatment with PBS, TNF, and IL-1β for various NF-κB pathway members and β-actin ($n = 3$). Data are presented as mean ± s.d. of $n = 3$ independent experiments. P, probability of no difference between cell lines by two-way ANOVA. ns, single, double, and triple asterisks (*, **, and ***): $P > 0.05$, $P < 0.05$, $P < 0.01$, and $P < 0.001$, respectively, for comparison of color-coded sh or p with control sh or p within each cell line by Bonferroni post-tests

it functions as an autocrine growth factor across tumor types. On the contrary, IL-1α/β selectively fostered MPE competence of *KRAS*-mutant carcinomas, in agreement with previous reports of IL-1β-induced NF-κB activation independent from IKKβ[55]. Our findings explain how the tumor microenvironment fuels tumor NF-κB activity[56] and link the pro-tumorigenic functions of IL-1β with *KRAS* mutations, setting a rationale for genotype-stratified future investigations on IL-1β functions and therapies in cancer.

Unbiased analyses identified cancer-elaborated CXCL1/PPBP, potent myeloid cell chemoattractants that drive inflammation and metastasis via CXCR1/CXCR2 on host cells[57,58], as the transcriptional targets of IL-1β-fostered *KRAS*-IKKα addiction. Indeed, *Cxcl1* expression was downregulated by *Kras* or *Chuk* silencing and IL-1β induced *Cxcl1* expression by two different cancer cell lines and *Ppbp* by LLC cells (MC38 cells do not express *Ppbp*[25]). Our experiments using CXCR1- and CXCR2-

deficient mice support that pleural tumor cell-secreted CXCL1/PPBP is cardinal for MPE and are in line with a previous study demonstrating increased production of CXCL1 by tumor cells during human MPE development that mobilizes regulatory T cells[59].

In addition to the mechanistic insights into host environment-fostered co-option of IKKα activity by mutant *KRAS*, our data bear therapeutic implications for KRAS inhibitors[39]. *KRAS* is notoriously undruggable, and proteasome and IKKβ inhibitors have yielded suboptimal results in mice and men with cancer. Focusing on lung cancer, a tumor with high *KRAS* mutation frequency[60], bortezomib has shown poor efficacy in clinical trials[61]. In animal models of lung cancer, bortezomib and IKKβ inhibitors caused resistance or paradoxical tumor promotion via development of secondary mutations, NF-κB inhibition in myeloid cells, or enhanced IL-1β secretion by tumor-associated

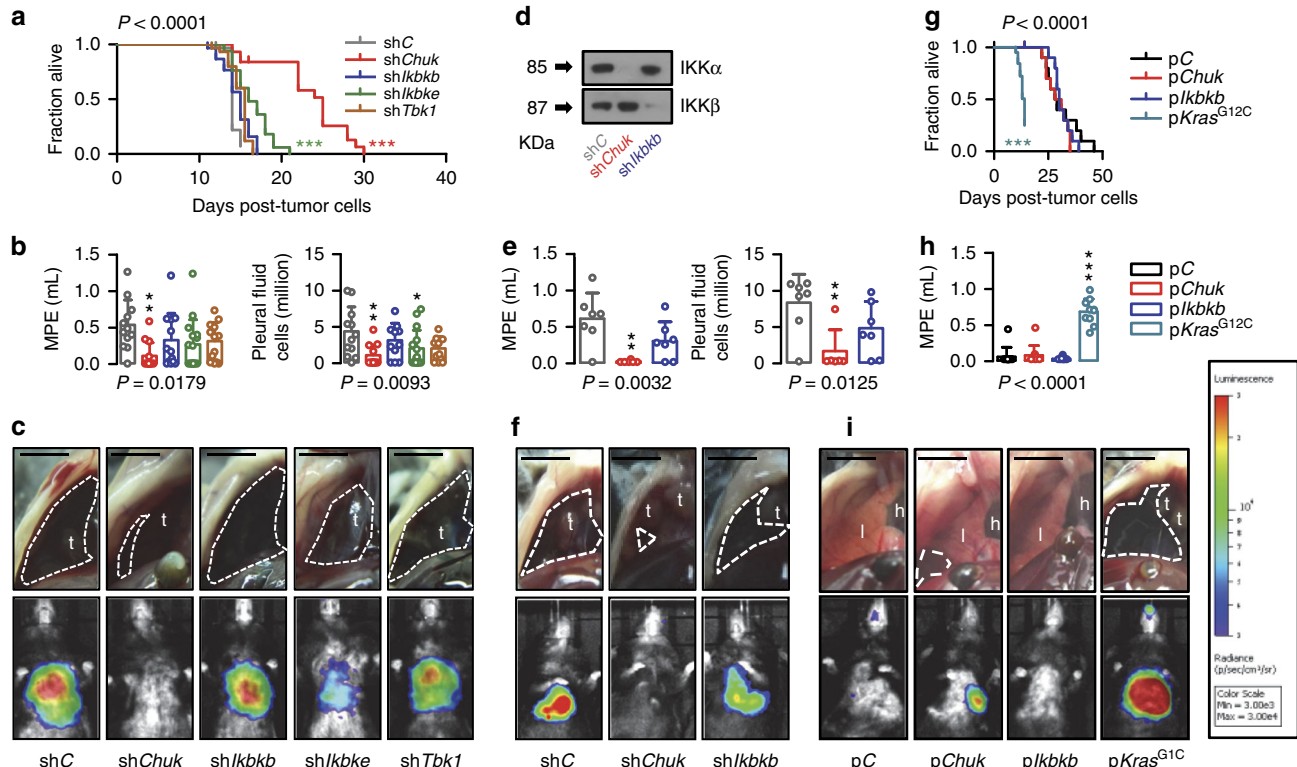

**Fig. 6** IKKα is required for mutant *KRAS*-induced malignant pleural effusion. **a–c** Malignant pleural disease induced by LLC cells (*Kras*$^{G12C}$) stably expressing p*NGL* NF-κB reporter and control shRNA (sh*C*) or shRNA targeting IKKα (sh*Chuk*), IKKβ (sh*Ikbkb*), IKKε (sh*Ikbke*), or TBK1 (sh*Tbk1*) transcripts (*n* is given in Table 3). Shown are Kaplan–Meier survival plot (**a**), data summaries of effusion volume and pleural fluid cells (**b**), and representative images of effusions (dashed lines) and pleural tumors (t) as well as bioluminescent images at day 13 after pleural injections of the indicated tumor cells (**c**). **d–f** Malignant pleural disease induced by MC38 cells (*Kras*$^{G13R}$) stably expressing p*NGL* NF-κB reporter and sh*C*, sh*Chuk*, or sh*Ikbkb* (*n* is given in Supplementary Table 3). Shown are immunoblots of cytoplasmic extracts (**d**), data summaries of effusion volume and pleural fluid cells (**e**), and representative images of effusions (dashed lines) and pleural tumors (t) as well as representative bioluminescent images at day 13 after pleural injections of the indicated tumor cells (**f**). **g–i** Malignant pleural disease induced by PANO2 cells (*Kras*$^{WT}$) stably expressing p*NGL* NF-κB reporter and control plasmid (sh*C*) or plasmid encoding IKKα (p*Chuk*), IKKβ (p*Ikbkb*), or mutant (p*Kras*$^{G12C}$) transcripts (*n* is given in Supplementary Table 3). Shown are Kaplan–Meier survival plot (**g**), data summary of effusion volume (**h**), and representative images of effusions (dashed lines) and pleural tumors (t), hearts (h), and lungs (l), as well as representative bioluminescent images at day 14 after pleural injections of the indicated tumor cells (**i**). Data are presented as mean ± s.d. *P*, probability of no difference between cell lines by overall log-rank test (**a**, **g**) or one-way ANOVA (**b**, **e**, **h**). ns, single, double, and triple asterisks (\*, \*\*, and \*\*\*): *P* > 0.05, *P* < 0.05, *P* < 0.01, and *P* < 0.001, respectively, for the indicated comparisons with control cells by Bonferroni post-tests. Scale bars, 0.5 cm

neutrophils through an unknown mechanism[15,16,53]. We show how *KRAS*-mutant cancer cells utilize myeloid-IL-1β in order to activate IKKα and alternative NF-κB signaling and to by-pass IKKβ canonical NF-κB dependence. We provide proof-of-concept data that *KRAS*-mutant cancer cells can be targeted by combined inhibition of KRAS and HSP90/IKKα/IKKβ signaling, a strategy that blocks IL-1β-inducible oncogenic NF-κB activation and in vivo MPE development, a cancer phenotype that requires mutant *KRAS*-potentiated, IL-1β-induced IKKα activity. These results challenge the prevailing focus on IKKβ for the development of anti-tumor drugs and establish IL-1β and IKKα as important targets in *KRAS*-mutant tumors.

In conclusion, we show that *KRAS*-mutant cancer cells use host IL-1β to sustain IKKα-mediated non-canonical NF-κB activity responsible for MPE development and primary drug resistance. We identify CXCL1/PPBP as effectors of MPE downstream of *KRAS*/IKKα addiction. Finally, we provide proof-of-concept data suggesting that *KRAS*/IKKα addiction may occur in human cancers and may be targeted by combined *KRAS*/IKKα inhibition.

## Methods

**Study approval**. All mouse experiments were prospectively approved by the Veterinary Administration of Western Greece (approval # 276134/14873/2) and

were conducted according to Directive 2010/63/EU (http://eur-lex.europa.eu/legal-content/EN/TXT/?uri=celex%3A32010L0063).

**Reagents**. D-Luciferin was from Gold Biotechnology (St. Louis, MO); lentiviral shRNA and puromycin from Santa Cruz (Dallas, TX); 3-(4,5-dimethylthiazol-2-yl)-2,5-diphenyltetrazolium bromide (MTT) assay and Hoechst 33528 from Sigma-Aldrich (St. Louis, MO); mouse gene ST2.0 microarrays and relevant reagents from Affymetrix (Santa Clara, CA); recombinant cytokines and growth factors from Immunotools (Friesoythe, Germany); NF-κB-binding ELISA from Active Motif (La Hulpe, Belgium); bortezomib, IMD-0354, 17-DMAG, and deltarasin from Sell-eckchem (Houston, TX); G418 from Applichem (Darmstadt, Germany); IL-1β and CXCL1 ELISA from Peprotech (London, UK); and primers from VBC Biotech (Vienna, Austria). Primers, antibodies, and lentiviral shRNA pools are listed in Supplementary Tables 6–8.

**Cells**. LLC, B16F10, PANO2, and A549 cells were from the National Cancer Institute Tumor Repository (Frederick, MD); MC38 cells were a gift from Dr. Barbara Fingleton (Vanderbilt University, Nashville, TN)[34,35], and AE17 cells from Dr. YC Gary Lee (University of Western Australia, Perth, Australia)[11,25]. All cell lines were cultured at 37 °C in 5% $CO_2$–95% air using Dulbecco's modified Eagle's medium (DMEM) containing 10% fetal bovine serum, 2 mM L-glutamine, 1 mM pyruvate, 100 U/mL penicillin, and 100 mg/mL streptomycin. Cell lines were tested annually for identity by short tandem repeats and for *Mycoplasma* Spp. by PCR. For in vivo injections, cells were harvested using trypsin, incubated with Trypan blue, counted in a hemocytometer, and 95% viable cells were injected intrapleurally[8,11,34,35].

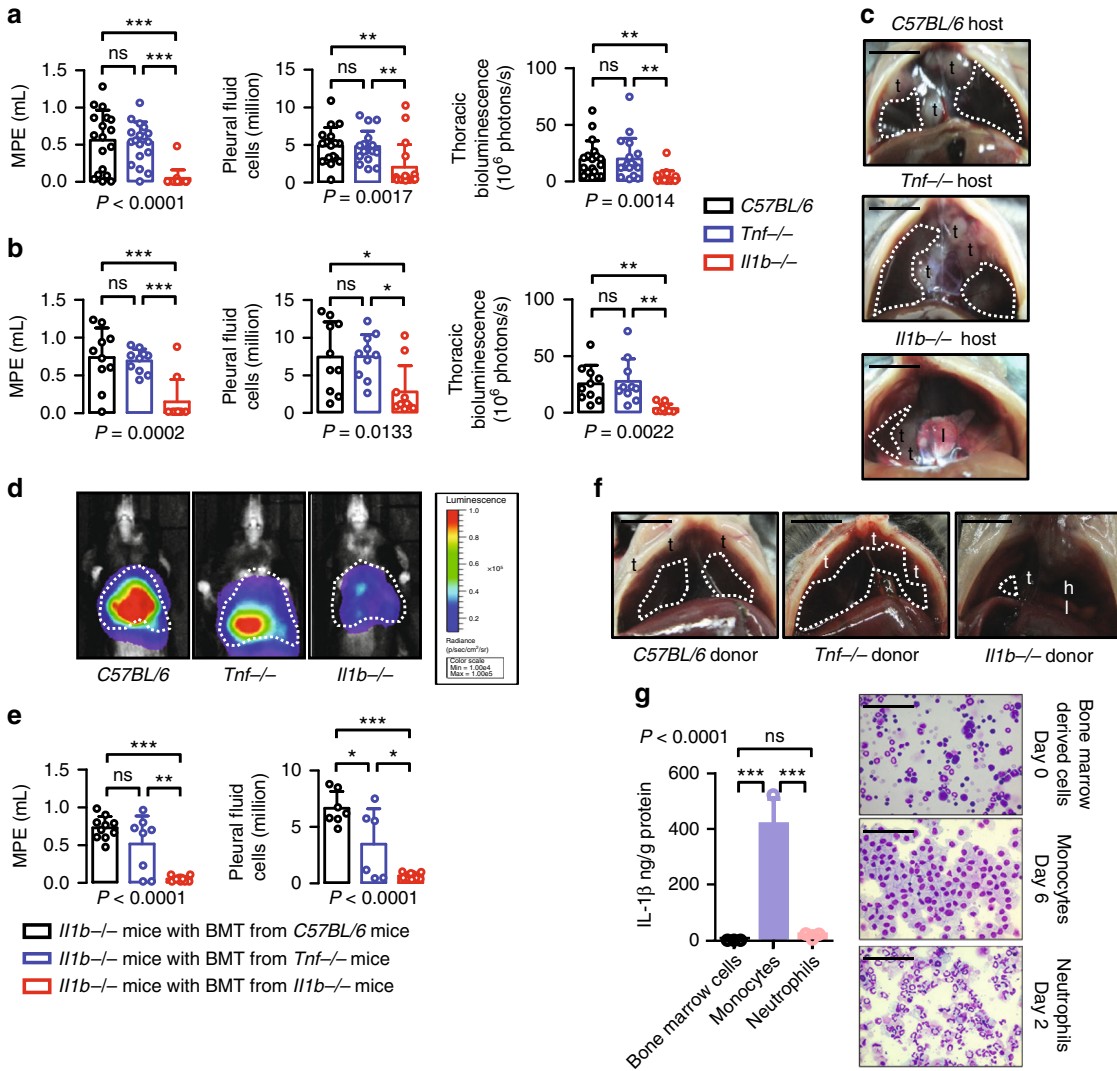

**Fig. 7** Myeloid cell-derived IL-1β drives mutant *KRAS*-IKKα addiction in malignant pleural effusion. **a** Malignant pleural disease induced by LLC cells (*Kras*[G12C]) stably expressing p*NGL* NF-κB reporter plasmid in wild-type *C57BL/6* mice (black) and TNF (blue) and IL-1β (red)-deficient mice (*Tnf–/–* and *Il1b–/–*, respectively; both *C57BL/6* background; *n* is given in Supplementary Table 3). Shown are data summaries of effusion volume, pleural fluid cells, and NF-κB-dependent thoracic bioluminescent signal. **b–d** Malignant pleural disease induced by MC38 cells (*Kras*[G13R]) stably expressing p*NGL* NF-κB reporter plasmid in wild-type *C57BL/6* (black), *Tnf–/–* (blue), and *Il1b–/–* (red) mice (all *C57BL/6* background; *n* is given in Supplementary Table 3). Shown are data summaries of effusion volume, pleural fluid cells, and NF-κB-dependent thoracic bioluminescent signal (**b**), representative images of effusions (dashed lines), pleural tumors (t), and lungs (l) (**c**), as well as representative bioluminescent images at day 13 after pleural injections of the indicated tumor cells (**d**). **e, f** Malignant pleural disease induced by LLC cells in *Il1b–/–* mice (*C57BL/6* background; *n* is given in Supplementary Table 3) that received total body irradiation (1100 Rad), same-day bone marrow transplants (10 million cells) from *C57BL/6* (black), *Tnf–/–* (blue), or *Il1b–/–* (red) donors, and pleural tumor cells after 1 month. Shown are data summaries of effusion volume and pleural fluid cells (**e**) and representative images of effusions (dashed lines), pleural tumors (t), lungs (l), and hearts (h) (**f**). **g** IL-1β protein secretion by *C57BL/6* mouse bone marrow-isolated myeloid cells 24 h after treatment with LLC supernatants; undifferentiated cells (day 0), neutrophils (day 2 after addition of 20 ng/ml G-CSF), and macrophages (day 6 after addition of 20 ng/ml M-CSF; *n* = 3 independent experiments). Data are presented as mean ± s.d. *P*, probability of no difference by one-way ANOVA. ns, single, double, and triple asterisks (*, **, and ***): *P* > 0.05, *P* < 0.05, *P* < 0.01, and *P* < 0.001, respectively, for the indicated comparisons by Bonferroni post-tests. Scale bars, 1 cm (**c, f**) and 100 μM (**g**)

**Mouse models and drug treatments**. *C57BL/6* (#000664), B6.129P2-*Cxcr1*[tm1Dgen]/J (*Cxcr1–/–*; #005820[36]), B6.129 S2(C)-*Cxcr2*[tm1Mwm]/J (*Cxcr2*[+/–]; #006848[37]), B6;129S-*Tnf*[tm1Gkl]/J (*Tnf–/–*; #003008[32]) (Jackson Laboratory, Bar Harbor, ME), and *Il1b*[tm1Yiw] (*Il1b–/–*; MGI #2157396[31]) mice were bred at the Center for Animal Models of Disease of the University of Patras. Male and female experimental mice and littermate controls were sex, weight (20–25 g), and age (6–12 weeks) matched. For MPE induction, mice received 150,000 cancer cells in 100 μL PBS intrapleurally. Mice were observed continuously till recovery and daily thereafter and were sacrificed when moribund (13–14 days post-tumor cells) for survival and pleural fluid analyses. Mice with pleural fluid volume ≥100 μL were judged to have a MPE and were subjected to pleural fluid aspiration, whereas animals with pleural fluid volume <100 μL were judged not to have a MPE and

were subjected to pleural lavage. Injection, harvest, and sample handling are described elsewhere[8–11,34,35]. Drug treatments were initiated 5 days post-tumor cells and consisted of daily intraperitoneal injections of 100 μL PBS containing no drug, deltarasin[39], 17-DMAG[28], or both at 15 mg/kg.

**Constructs**. p*NGL*, p*IκBα*DN, and p*CAG.LUC* (#74409) have been described elsewhere[8,25,33]. Lentiviral shRNA pools (Santa Cruz) are described in Supplementary Table 8. A p*MIGR1*-based (#27490) bicistronic retroviral expression vector was generated by replacing *eGFP* sequences with puromycin resistance gene (#58250). *Kras*[G12C],*Chuk*, *Ikbkb*, *Ikbke*, and *Tbk1* cDNAs were cloned via reverse transcriptase-PCR (RT-PCR) from LLC or MC38 RNA using specific primers

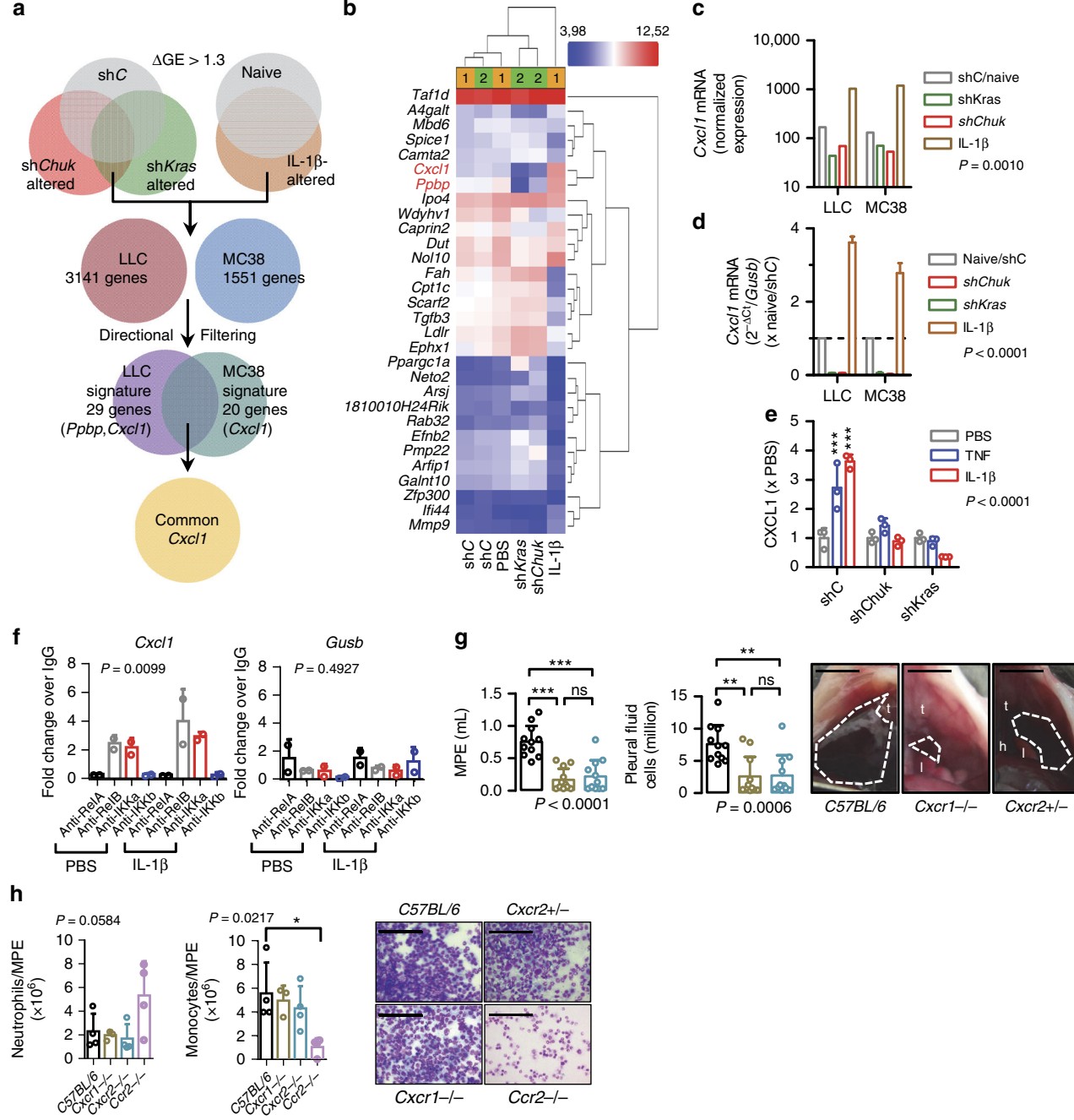

**Fig. 8** CXCL1/PPBP are the downstream effectors of KRAS/IL-1β/IKKα signaling in malignant pleural effusion. **a–c** LLC and MC38 cells were stably transfected with sh*C* or sh*Kras* or shChuk or were stimulated with 1 nM IL-1β for 4 h, and total cellular RNA was examined by Affymetrix mouse gene ST2.0 microarrays. **a** Venn diagram of analytic strategy employed: transcripts altered >1.3-fold in one direction by sh*Kras* and sh*Chuk* and in the other by IL-1β were filtered for each cell line and are given in Supplementary Tables S1–S2. These gene sets, coined KRAS/IL-1β/IKKα signatures, were crossexamined and only *Cxcl1* was common to both. **b** Unsupervised hierarchical clustering of LLC cell results by the 29-gene KRAS/IL-1β/IKKα signature accurately clustered three control samples together, sh*Kras* and sh*Chuk* samples together, and IL-1β-stimulated cells apart. **c** *Cxcl1* mRNA normalized expression levels by microarray (n = 2 independent experiments). **d** *Cxcl1* mRNA expression by qPCR relative to *Gusb* (n = 3 independent experiments). **e** CXCL1 protein secretion by LLC cells stably expressing sh*C*, shChuk, or sh*Kras* after 24 h of stimulation with PBS or 1 nM TNF or IL-1β (n = 3 independent experiments). **f** Chromatin immunoprecipitation (ChIP) was performed in PBS- or IL-1β-treated LLC cells, followed by immunoprecipitation with the indicated antibodies. The immunoprecipitates were then detected by qPCR. Data are shown as fold enrichment of *Cxcl1* or *Gusb* promoter in each antibody immunoprecipitate over control IgG immunoprecipitate. **g** Malignant pleural disease induced by LLC cells in *C57BL/6*, *Cxcr*[+/−], and *Cxcr2*[+/−] mice (n is given in Supplementary Table 3). Shown are data summaries of effusion volume and pleural fluid cells, as well as representative images of effusions (dashed lines), pleural tumors (t), lungs (l), and hearts (h). **h** Data summaries of *C57BL/6*, *Cxcr1*[−/−], *Cxcr2*[+/−] and *Ccr2*[−/−] pleural neutrophils and monocytes, accompanied by microphotographs. Data are presented as mean ± s.d. P, probability of no difference by two-way (**c–e**) or one-way (**f–h**) ANOVA. ns, single, double, and triple asterisks (** and ***): P > 0.05, P < 0.05, P < 0.01, and P < 0.001, respectively, for comparison with PBS (**e**) or indicated (**g**, **h**) by Bonferroni post-tests. Scale bars 1 cm (**g**) and 200 μM (**h**)

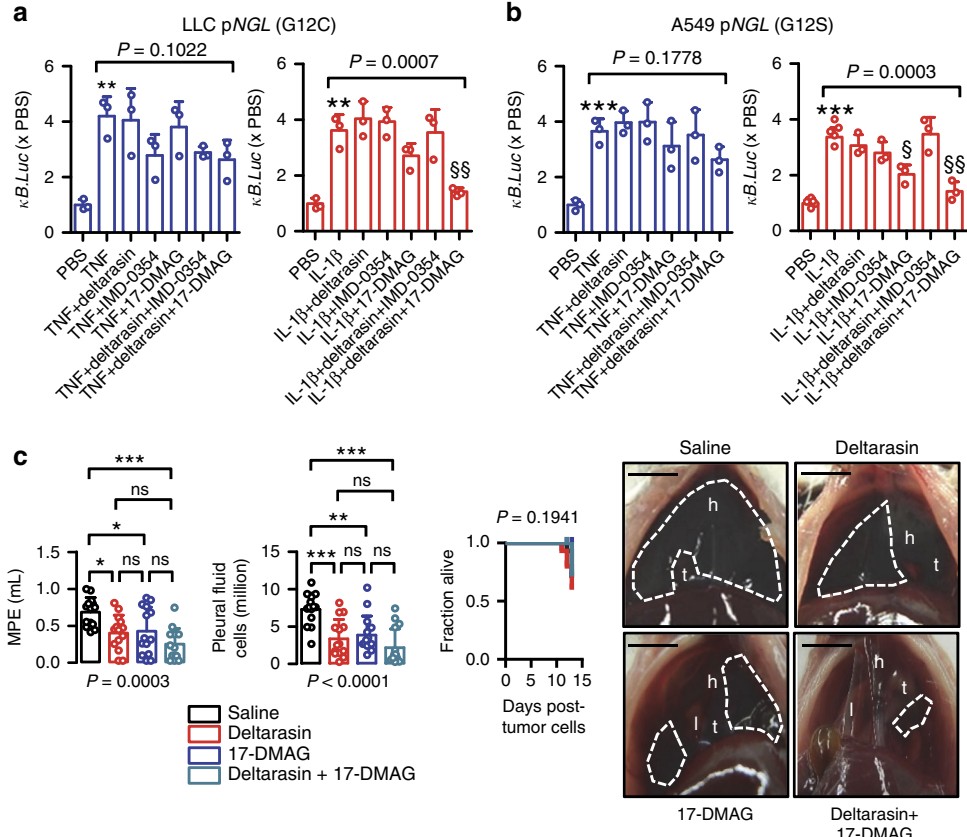

**Fig. 9** Combined targeting of mutant *KRAS* and IKKα abolishes IL-1β-induced NF-κB activation and malignant pleural effusion development. **a, b** Bioluminescent detection of NF-κB reporter activity in LLC (**a**; *C57BL/6* Lewis lung carcinoma, *Kras*[G12C]) and A549 (**b**; human lung adenocarcinoma, *Kras*[G12S]) cells stably expressing p*NGL* under PBS or 1 nM TNF- or IL-1β-stimulated conditions (4 h), with or without pretreatment with 1 μM deltarasin, IMD-0354, or 17-DMAG alone or in combination (*n* = 3 independent experiments). Note four-fold induction of NF-κB reporter activity by both TNF and IL-1β. Note also inability of any treatment to block TNF-induced NF-κB activation and of any standalone treatment except 17-DMAG to inhibit IL-1β-induced NF-κB activation. Finally, note complete abrogation of IL-1β-induced NF-κB activation in both cell lines by deltarasin/17-DMAG combination. Data are presented as mean ± s.d. *P*, probability of no difference by one-way ANOVA (PBS group excluded). Double and triple asterisks (\*\* and \*\*\*): *P* < 0.01 and *P* < 0.001, respectively, for comparison with PBS by Student's *t*-tests. Single and double section symbols (§ and §§): *P* < 0.05 and *P* < 0.01, respectively, for comparison with TNF or IL-1β by Bonferroni post-tests. **c** Malignant pleural disease induced by LLC cells in wild-type *C57BL/6* mice treated with deltarasin and/or 17-DMAG. Mice received pleural LLC cells, were allowed 5 days for pleural tumor development, and were randomized to daily intraperitoneal treatments with saline (100 μL), deltarasin, 17-DMAG, or both (both at 15 mg/Kg in 100 μL saline; *n* is given in Supplementary Table 3). Shown are data summaries of effusion volume and pleural fluid cells and Kaplan–Meier survival plot, as well as representative images of effusions (dashed lines), pleural tumors (t), lungs (l), and hearts (h). Data are presented as mean ± s.d. *P*, probability of no difference by one-way ANOVA or log-rank test. ns, single, double, and triple (\*, \*\*, and \*\*\*): *P* > 0.05, *P* < 0.05, *P* < 0.01, and *P* < 0.001, respectively, for the indicated comparisons by Bonferroni post-tests. Scale bars, 1 cm

(Supplementary Table 6) and were subcloned into peGFP-C1 (Takara, Mountain View, CA). *eGFP*, *eGFP.Kras*[G12C], *eGFP.Chuk*, *eGFP.Ikbkb*, *eGFP.Ikbke*, and *eGFP. Tbk1* cDNAs were subcloned into the new retroviral expression vector (#58249, #64372,# 87033, #58251, #87444, and #87443, respectively). Retroviral particles were obtained by co-transfecting HEK293T cells with retroviral vectors, *pMD2.G* (#12259), and *pCMV-Gag-Pol* (Cell Biolabs, San Diego, CA) at 1.5:1:1 stoichiometry using CaCl$_2$/BES. After 2 days, culture media were collected and applied to cancer cells. After 48 h, media were replaced by selection medium containing 2–10 μg/mL puromycin. Stable clones were selected and subcultured[11]. For stable plasmid/shRNA transfection, 10$^5$ tumor cells in six-well culture vessels were transfected with 5 μg DNA using Xfect (Takara), and clones were selected by G418 (400–800 μg/mL) or puromycin (2–10 μg/mL).

**Cellular assays.** In vitro cancer cell proliferation was determined using MTT assay. Nuclear extracts were assayed for *RelA*, *RelB*, *c-Rel*, P50, and P52 DNA-binding activity using a commercially available ELISA kit (Transam, Active Motif, Belgium). All cellular experiments were independently repeated at least thrice.

**Bioluminescence imaging.** Living cells and mice were imaged 0, 4, 8, 24, and 48 h after cellular treatments and 0 h, 4 h, and 12–14 days after pleural delivery of p*NGL*-expressing cells on a Xenogen Lumina II (Perkin-Elmer, Waltham, MA)

after addition of 300 μg/mL D-luciferin to culture media or isoflurane anesthesia and delivery of 1 mg intravenous D-luciferin to the retro-orbital veins[8–11,16,25,34,35]. Data were analyzed using Living Image v.4.2 (Perkin-Elmer).

**qPCR and microarray.** RNA was isolated using Trizol (Invitrogen, Carlsbad, CA) and RNAeasy (Qiagen, Hilden, Germany) was reverse transcribed using Superscript III (Invitrogen), and RT-PCR or qPCR was performed using SYBR Green Master Mix in a StepOnePlus (Applied Biosystems, Carlsbad, CA) and specific primers (Supplementary Table 6). Ct values from triplicate qPCR reactions were analyzed by the 2$^{-\Delta\Delta CT}$ method[62] relative to *Gusb* mRNA levels. For microarray, RNA was extracted from triplicate cultures of 10$^6$ cells. Five micrograms pooled total RNA were quality tested on an ABI 2000 (Agilent Technologies, Sta. Clara, CA), labeled, and hybridized to GeneChip Mouse Gene 2.0 ST arrays (Affymetrix, St. Clara, CA). For analysis of differential gene expression (ΔGE) and unsupervised hierarchical clustering, Affymetrix Expression and Transcriptome Analysis Consoles were used.

**Chromatin immunoprecipitation.** LLC cells were treated with PBS or 1 nM IL-1β, and 30 min later, cells were fixed sequentially with 2 mM di(N-succinimidyl) glutarate (Sigma) and 1% formaldehyde (Sigma) and quenched with 0.125 M glycine, followed by lysis with 1% sodium dodecyl sulfate (SDS), 10 mM EDTA,

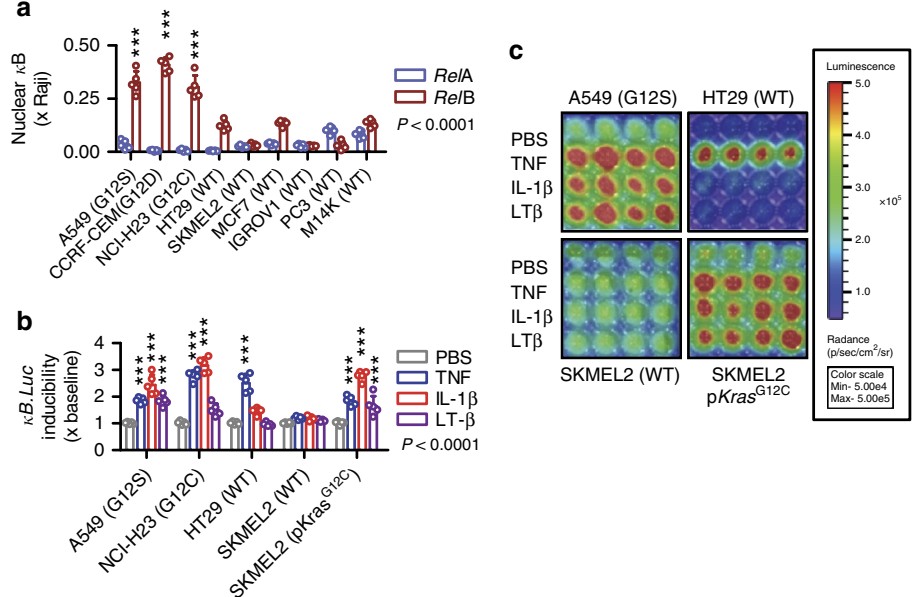

**Fig. 10** Non-canonical endogenous and IL-1β-inducible NF-κB activation of *KRAS*-mutant human tumor cells. Different human cancer cell lines with (*KRAS*^MUT: A549, *KRAS*^G12S; CCRF-CEM, *KRAS*^G12D; NCI-H23, *KRAS*^G12C) or without (*KRAS*^WT; HT29, SKMEL2, MCF7, IGROV1, PC3, and M14K) *KRAS* mutations were assessed for NF-κB activation at resting and stimulated conditions in vitro. **a** Data summary (*n* = 5 independent experiments) of DNA NF-κB motif-binding activity of nuclear extracts by NF-κB ELISA relative to nuclear extracts of Raji leukemia cells. Note increased nuclear *Rel*B and P52 binding activity of *KRAS*^MUT compared with *KRAS*^WT cells. **b, c** Bioluminescent detection of NF-κB reporter activity in A549, NCI-H23, HT29, and SKMEL2 cells stably expressing p*NGL*, as well as in SKMEL2 cells stably expressing p*NGL* and p*Kras*^G12C (**b**; data summary of *n* = 5 independent experiments; **c**: representative bioluminescent images) during 4-h incubation with PBS or 1 nM TNF, IL-1β, or lymphotoxin (LT)-β. Note IL-1β-induced NF-κB activity of *KRAS*^MUT but not of *KRAS*^WT cells. Note also instalment of IL-1β-induced NF-κB activation in SKMEL2 cells by p*Kras*^G12C (SKMEL2 cells expressing p*C* behaved exactly as parental cells). Data are presented as mean ± s.d. *P*, probability of no difference by two-way ANOVA. Triple astrisks (***): *P* < 0.001 for comparison with *Rel*A (**a**) or PBS (**b**) by Bonferroni post-tests

and 50 mM Tris pH 8. Sonication was performed in a Bioruptor (Diagenode) for 40 cycles (30 s on/off) power settings high), using $3 \times 10^6$ cells; 20 μg of chromatin was precipitated with 5 μg of *Rel*A, *Rel*B, IKKα, or IKKβ antibody or a mouse control immunoglobulin G (IgG). Immunoprecipitates were retrieved with 50 μl of magnetic Dynabeads conjugated to protein G (Invitrogen) and subjected to quantitative real-time PCR (Applied Biosystems StepOne), using the Kapa SYBR Fast qPCR Kit (KapaBiosystems, KK4605) for amplification of the *Cxcl1* promoter or *Gusb* as control. The sequences of the primers used for Cxcl1 promoter are: 5′-ATA-CAGCAGGGTAGGGATGC, 3′-TTGCCAACTGTTTTTGTGG. The sequences of the primers used for Gusb are: 5′-TTACTTTAAGACGCTGATCACC, 3′-ACCTCCAAATGCCCATAGTC.

**BM cell derivation and transfer**. For adoptive BM replacement, *Il1β*−/− mice (*C57BL/6* background) received 10 million BM cells flushed from the femurs and tibias of *C57BL/6*, *Tnf*−/−, or *Il1β*−/− donors (*C57BL/6* background) intravenously 12 h after total-body irradiation (1100 Rad)[11,25,34,35]. One mouse in each experiment was not engrafted (sentinel) and was observed till moribund between days 5 and 15 post-irradiation. The mice were left to be engrafted for 1 month, when full BM reconstitution is complete, before experimental induction of pleural carcinomatosis via intrapleural injection of LLC cells. For BM cell retrieval, BM cells were flushed from *C57BL/6* femurs and tibias using full DMEM and were simply cultured in full culture media (the same used for cancer cell line cultures), supplemented with 20 ng/ml M-CSF or G-CSF in order for cells to differentiate to monocytes or neutrophils, respectively. Supernatants and cytocentrifugal specimens were obtained at day 0 for undifferentiated cells, day 2 for neutrophils, and at day 6 for monocytes/macrophages.

**Immunoblotting**. Nuclear and cytoplasmic extracts were prepared using the NE-PER Extraction Kit (Thermo, Waltham, MA), separated by 12% SDS polyacrylamide gel electrophoresis, and electroblotted to polyvinylidene difluoride membranes (Merck Millipore, Darmstadt, Germany). Membranes were probed with specific antibodies (Supplementary Table 7) and were visualized by film exposure after incubation with enhanced chemiluminescence substrate (Merck Millipore, Darmstadt, Germany).

**Electrophoretic mobility shift assay (EMSA)**. Nuclear extracts were prepared using the NE-PER Extraction Kit. Proteins (10 μg) were incubated with NF-κB biotin-labeled probe using a commercially available non-radioactive EMSA Kit

(Signosis Inc, Santa Clara, USA). DNA–protein complexes were electrophoresed in a prerinsed 6.5% polyacrylamide gel, transferred to a positively charged nylon membrane, and were visualized by film exposure after incubation with enhanced chemiluminescence substrate. For gel shift reactions, proteins were incubated with the specific antibody for 1 h at 4 °C before probe incubation. The antibodies used for observing the supershifted bands were *Rel*A and *Rel*B. IgG antibody served as negative control for super-shift assays.

**Immunofluorescence**. For immunofluorescence, cells were fixed in 4% paraformaldehyde overnight at 4 °C and were labeled with the indicated primary antibodies (Supplementary Table 7) followed by incubation with fluorescent secondary antibodies (Invitrogen, Waltham, MA; Supplementary Table 7). Cells were then counterstained with Hoechst 33258 (Sigma-Aldrich, St. Louis, MO) and mounted with Mowiol 4-88 (Calbiochem, Gibbstown, NJ). For isotype control, the primary antibody was omitted. Fluorescent microscopy was carried out on an AxioObserver.D1 inverted microscope (Zeiss, Jena, Germany) connected to an AxioCam ERc 5 s camera (Zeiss), and digital images were processed with the Fiji academic imaging freeware[63].

**Statistics**. Sample size was calculated using G*power (http://www.gpower.hhu.de/)[64] assuming $\alpha = 0.05$, $\beta = 0.05$, and $d = 1.5$, tailored to detect 30% differences between means with 20–30% SD spans, yielding $n = 13$/group. Animals were allocated to groups by alternation (treatments or cells) or case–control-wise (transgenic animals). Data acquisition was blinded on samples coded by non-blinded investigators. No data were excluded. All data were examined for normality by Kolmogorov–Smirnof test and were normally distributed. Values are given as mean ± SD. Sample size (*n*) refers to biological replicates. Differences in means were examined by *t*-test and one-way or two-way ANOVA with Bonferroni post-tests, in frequencies by Fischer's exact or $\chi^2$ tests, and in Kaplan–Meier survival estimates by log-rank test, as appropriate. *P*-values are two-tailed. *P* < 0.05 was considered significant. Analyses and plots were done on Prism v5.0 (GraphPad, La Jolla, CA).

**Data availability**. All new plasmids have been deposited at the Addgene plasmid repository (https://www.addgene.org/search/advanced/?q=stathopoulos) and their IDs (#) are given in the text. Microarray data are available at the GEO (http://www.ncbi.nlm.nih.gov/geo/; Accession IDs: GSE93369 and GSE93370). The authors declare that all the other data supporting the findings of this study are available

within the article and its supplementary information files and from the corresponding authors upon reasonable request.

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

## Acknowledgements

This work was supported by European Research Council 2010 Starting Independent Investigator and 2015 Proof of Concept Grants (260524 and 679345, to GTS), by European Respiratory Society 2013 Romain Pauwels Research Award (to G.T.S.), by a Hellenic Association for Molecular Cancer Research Award 2015 (to A.M.), by a Hellenic Thoracic Society Research Award 2014 (to M.V.), and by an Immunotools Award 2014 (to A.D.G.). The authors thank the University of Patras Center for Animal Models of Disease for experimental support.

## Author contributions

A.M. designed and performed NF-κB ELISA, immunoblotting, EMSA, drug testing, transfections, reporter assays, and most in vivo experiments, quantified and analyzed the data, provided critical intellectual input, and wrote the paper draft; I.L. isolated BMMCs; M.V. designed and performed reporter assays and in vivo experiments including bioluminescent imaging, quantified and analyzed the data, and provided critical intellectual input; H.A. and A.D.G. performed pNGL induction studies, mutant KRAS and IKK silencing/overexpression and relevant in vitro assays, and drug testing; A.K. performed CHIP experiments; I.G. did qPCR experiments; G.A.G. and A.C.K. performed in vivo deltarasin/17-DMAG treatment experiments; M.I. did pleural fluid cell counts; N.I.K. analyzed microarray; T.A. cloned eukaryotic expression vectors; C.J.-P. performed NF-κB ELISA; Y.I. provided analytical tools and critical intellectual input; D.K. performed total body irradiation; T.S.B. provided pNGL and critical intellectual input; S.T. provided analytical tools and critical intellectual input; M.S. performed immunofluorescence; G.T. S. conceived the idea and supervised the study, designed experiments, analyzed the data, wrote the paper, and is the guarantor of the study's integrity. All authors reviewed, edited, and concur with the submitted manuscript.

## Additional information

**Competing interests:** The authors declare no competing financial interests.

