## [Peer Review File · Nature Communications]

Reviewers' comments:

Reviewer #1:

(Remarks to the Author):

The authors of the manuscript entitled: "Malignant pleural effusion (MPE) is a frequent metastatic manifestation of human cancers," report the study of the underlying mechanism of mutant KRAS driven MPE. MPE describes the condition in which cancer causes an abnormal amount of fluid to collect between the thin layers of tissue (pleura) lining the outside of the lung and the wall of the chest cavity. Lung cancer and breast cancer account for about 50-65% of MPE. In addition, MPE also occurs frequently in pleural mesothelioma and lymphoma. The study described by Marazioti et al. showed that IKK α -mediated non-canonical NF- κ B activation of KRAS-mutant tumor cells mediates MPE development, which is further stimulated by host-provided IL-1 β , resulting in increased CXCL1 secretion that promotes MPE associated inflammation. This is the main point of this manuscript. Although the idea has certain novelty, at this moment, additional experiments are required for validation. The authors suggest that mutant KRAS induced expression of IL-1 β that activates NF- κ B to induce MPE. While the mutant KRAS signal cascade described by the authors may be a key branch of KRAS downstream effector pathways involved in tumorigenesis and metastasis, the detailed mechanistic study is lacking. There are several questions about the experimental designs and approaches. Furthermore, additional validation experiments are needed to strengthen their finding. The specific comments are shown as follows.

1. Figure 1A, the authors stated: "Overall NF- κ B activity is not correlated with KRAS mutation." Please explain. In Figure 1A, why there was no bioluminescence activity in LLC(G12C)? To validate the results, isogenic WT cell line with and without mutant KRAS should be established to examine NF κ B activation by mutant KRAS.
2. In Figure 1D, these immunoblotting experiments are lacking of the control for the quality of nuclear and cytosol extracts. B16F10(WT) cells has a strong RelB band, suggesting non-canonical NF κ B activation, the results are contradictory to what was stated.
3. Why easy, simple, and reliable EMSA was not used to examine NF- κ B activation? EMSA approach should be additional experiments to validate their finding, which is applied to Figure 4K and 5E.
4. In Figure 2, how KRAS mutant cells induce or express IL-1beta expression is a key question, but it is unanswered.
5. In Figure 3, it is still unclear whether IL-1beta activates non-canonical NF κ B pathway only. Both canonical and non-canonical NF- κ B activation should be examined carefully and demonstrate activation of anyone or both pathways.
6. It is unclear there is any difference in the source of IL-1beta that would make much differences.
7. In Figure 8, the authors show that CXCL1/PPBP is downstream effector of mutant KRAS in induction of MPE. These results may suggest CXCL1/PPBP is required for inducing MPE, but it is not clear whether it is sufficient to induce MPE.
8. In Figure 9, the survival curve should be shown. Is the inhibitor such as deltarsan very toxic?
9. For Figure 10, authors proposed that tumor cells without KRAS mutations respond to host cytokines by IKK β -mediated canonical NF- κ B activation resulting in pleural carcinomatosis without malignant pleural effusion (MPE), and mutant KRAS cancer cells respond to myeloid IL-1 β in addition to cytokines by IKK α -mediated non-canonical NF- κ B activation leading to in CXCL1/PPBP-effected pleural carcinomatosis with MPE. If so, what is the mechanism? What is the evidence to support such model?

Reviewer #2:

(Remarks to the Author):

The investigators have previously shown that KRAS mutations are associated with malignant pleural effusions (MPE). In this paper, they have shown that this is dependent on IL-1 signaling that is transduced through IKK α at least in part through increasing CXCL1 gene expression. Their data show that this MPE formation is relatively drug resistant but that concomitant blocking of KRAS and IKK α effectively ameliorates the development of MPE. The work is detailed, comprehensive and using innovative methods. Although some MPE are alleviated by chemotherapy many patients suffer from MPE especially those with limited lung function or excessively large effusions. MPE are notoriously difficult to manage and current management is dependent on palliation and placement of an indwelling tube thoracotomy in patients during their last few weeks of life. The translational application of combination KRAS and IKK α treatment could greatly improve the quality and quantity of the life of patients with terminal cancer with MPE. Although the work is comprehensive, there are some minor concerns that should be addressed.

1) The introduction indicates that mutant KRAS exerts its pro-MPE effect through CCL2, but the new data show that the effect is exerted through CXCL1 gene expression and is dependent on myeloid cell recruitment. This should be resolved since CCL2 would be expected to recruit monocytes whereas CXCL1 would be expected to recruit neutrophils. The data is ambiguous as to whether monocytes or neutrophils are recruited in the MPE associated with KRAS mutations.

2) In conjunction with the comments above, it would be interesting to know whether monocytes or neutrophils or both are the source of the endogenous IL-1. This is an important point since monocytes produce much more IL-1 per cell than do neutrophils.

3) Although the paper refers to previous references that use the NGL plasmid, I think that this should be better defined, including showing the sequence code of the NF- κ B binding motif, since so much of the data was acquired with this reporter vector.

4) In addition to showing expression of the reported plasmid as a surrogate marker of the NF- κ B activation pathway, showing binding of RelB/p52 binding to the enhancer region of the IL-1R, by EMSA and ChIP, would strengthen the data set. This would be useful to support the conclusion that KRAS mutation enhances the responsiveness to IL-1 by increasing IL-1R gene expression.

5) The western blots for I κ B α shown in figure 4 is of poor quality and in Figure 5 it appears that immunoreactive I κ B α is depleted by treatment with shCHuK and shI κ B α which is not expected and should be explained.

6) Similar to the point made above, showing binding of RelB/p52 binding to the enhancer region of the CXCL1, by EMSA and ChIP, would strengthen the data set.

7) As pointed out by the investigators, IKK α could serve an epigenetic function through regulating post-translational modifications of histone 3. This could result in enhancement of IL-1R or CXCL1 gene expression and assisted by a histone ChIP. Although this might not be within the scope of this manuscript, the discussion could be elaborated that the combination of the KRAS mutation and epigenetic influences of IKK α in addition to the non-canonical activation of NF- κ B could influence MPE formation.

8) The cartoon shown as Figure 10D is very nice but the labels could be enlarged relative to the drawings.

RESPONSE TO REVIEWERS' COMMENTS

REVIEWER #1 COMMENTS

“The authors of the manuscript entitled: “Malignant pleural effusion (MPE) is a frequent metastatic manifestation of human cancers,” report the study of the underlying mechanism of mutant KRAS driven MPE. MPE describes the condition in which cancer causes an abnormal amount of fluid to collect between the thin layers of tissue (pleura) lining the outside of the lung and the wall of the chest cavity. Lung cancer and breast cancer account for about 50-65% of MPE. In addition, MPE also occurs frequently in pleural mesothelioma and lymphoma. The study described by Marazioti et al. showed that IKK α -mediated non-canonical NF- κ B activation of KRAS-mutant tumor cells mediates MPE development, which is further stimulated by host-provided IL-1 β , resulting in increased CXCL1 secretion that promotes MPE associated inflammation. This is the main point of this manuscript. Although the idea has certain novelty, at this moment, additional experiments are required for validation. The authors suggest that mutant KRAS induced expression of IL-1 β that activates NF- κ B to induce MPE. While the mutant KRAS signal cascade described by the authors may be a key branch of KRAS downstream effector pathways involved in tumorigenesis and metastasis, the detailed mechanistic study is lacking. There are several questions about the experimental designs and approaches. Furthermore, additional validation experiments are needed to strengthen their finding. The specific comments are shown as follows.”

We sincerely thank the reviewer for his/her positive appraisal of our work. We have carefully revised the manuscript so as to address all of his/her comments.

1. **“Figure 1A, the authors stated: “Overall NF-κB activity is not correlated with KRAS mutation.” Please explain. In Figure 1A, why there was no bioluminescence activity in LLC(G12C)? To validate the results, isogenic WT cell line with and without mutant KRAS should be established to examine NF-κB activation by mutant KRAS.”**

We thank the reviewer for his/her comments. In Figure 1A (in revised manuscript is Figure 1B) our results show that unstimulated NF-κB activity did not segregate by *KRAS* mutation status. B16F10 cell line which is *Kras*^{WT} show a high *κB.Luc* activity that corresponds to canonical activity (as depicted in Figure 1C), whereas LLC cell line which is *Kras*^{MUT} has lower overall NF-κB activity that is mainly non canonical. Therefore, the presence of mutant *KRAS* in a cancer cell line does not set its overall NF-κB activation but determines which pathway, canonical or non canonical, is activated. In Figure 1B (revised manuscript) LLC cells present bioluminescent activation in all 3 experiments and is summarized in the relative graph. In Figure 1C we incorporated data from an isogenic *Kras*^{WT} cell line PANO2p*NGL*, with very low NF-κB activity, where we examined NF-κB activation after transient transfection with pC or p*Kras*^{G12C}. Indeed, the presence of *KRAS* mutation led to statistically increased overall NF-κB activity.

2. **“In Figure 1D, these immunoblotting experiments are lacking of the control for the quality of nuclear and cytosol extracts. B16F10(WT) cells has a strong RelB band, suggesting non-canonical NFκB activation, the results are contradictory to what was stated.”**

We appreciate the reviewers' comment. We used the commercially available NE-PER extraction kit (Thermo Scientific) for preparing our immunoblotting samples as it isolates high quality cytoplasmic and nuclear extracts from mammalian cells and efficiently solubilize and separate cytoplasmic and nuclear proteins into fractions with minimal cross-contamination.

We totally agree that *RelB* blots for all five cell lines (Figure 1F in the revised manuscript) seems not to be in agreement with the other results. We repeated this experiment and substituted the cytoplasmic and nuclear *RelB* blots as the new results were quite different and more reasonable (cytoplasmic and nuclear *RelB* band was absent for almost all cell lines, an unreasonable result that probably was due to technical issues).

3. **“Why easy, simple, and reliable EMSA was not used to examine NF-κB activation? EMSA approach should be additional experiments to validate their finding, which is applied to Figure 4K and 5E.”**

We appreciate the reviewers' comment. In our laboratory we have established four different and quite modern techniques, broadly used for NF-κB activation by the scientific community, Transam NF-κB Elisa kit, Western blot analysis, Immunofluorescence and a method based on luciferase placed under the control of a promoter containing the NF-κB consensus sequence. These techniques are modern, combine fast and user-friendly formats, with high sensitivity and specificity for NF-κB subunits activation. However, due to the Reviewer's

suggestion we incorporated data of EMSA experiments in Figure 4 (Fig.4L). We used the Signosis EMSA kit that is a non- radioactive EMSA-NF- κ B kit since we don't have an official approval or the appropriated facilities to perform radio-active assays.

- 4. “In Figure 2, how KRAS mutant cells induce or express IL-1beta expression is a key question, but it is unanswered.”**

We appreciate the reviewers' comment. We explain in more detail in the Results section that *IL1a/IL1b* expression was infinitely small in all cell lines, which implies that *KRAS* mutant cells do not actually express IL-1beta. This result is clearly shown in Figure 2D as the axis range in qPCR results is too small for *IL1b* compared with the relative axis range of *IL1r1* graph. Moreover, in Figure 7G where we performed IL-1 β ELISA we didn't observe any cytokine secretion in LLC cell - free supernatants (the first column represents IL-1 β secretion in supernatants from undifferentiated myeloid cells treated with LLC supernatants and the result shows protein levels to the lower detection limit).

- 5. “In Figure 3, it is still unclear whether IL-1beta activates non-canonical NFkB pathway only. Both canonical and non-canonical NF-kB activation should be examined carefully and demonstrate activation of anyone or both pathways.”**

We appreciate the reviewers' comment. Indeed, using the pNGL plasmid we only observe the overall NF- κ B activity and we certainly cannot distinguish if canonical or non- canonical pathway is activated or inhibited. However, by using commercially available NF- κ B inhibitors we can presume the pathway that leads to luciferase induction. In order to see which pathway IL-1 β activates we pretreated *KRAS* mutant pNGL LLC and A549 lung adenocarcinoma cells (mouse and human respectively) with the IKK β inhibitor IMD-0354 or with the dual IKK β /IKK α inhibitor 17-DMAG. The results clearly show that the canonical pathway inhibitor IMD-0354 cannot reduce at all the luciferase signal in LLC cells after IL-1 β exposure, implying that IL-1 β activates the non-canonical pathway in these cells. These data are shown in Figure 9A.

- 6. “It is unclear there is any difference in the source of IL-1beta that would make much differences.”**

We thank the reviewer for this comment. IL-1 β is a potent pro-inflammatory cytokine that is present in pleural space during MPE development. It is produced by various cell types of innate immune cells. In the pleural space the inflammatory cells that are usually recruited are monocytes, neutrophils and mast cells. We have previously shown that mast cells secrete IL-1 β in response to the presence of *KRAS* mutant pleural tumors (Giannou *et al* 2015 JCI; Reference 35). Here we provide evidence that the main source of IL-1 β secretion in the pleural space during MPE development is monocytes (Figure 7G). Moreover, neutrophils, to a much lesser extent than monocytes, also secreted IL-1 β cytokine in response to *KRAS* mutant pleural tumors. Collectively, we can state

that irrespective of which immune cell type is the IL-1 β source during MPE development the mechanism of its secretion as well as its effects in tumor cells is pretty much the same, i.e. leads *KRAS* mutant tumors to increased non-canonical NF- κ B activity.

7. **“In Figure 8, the authors show that CXCL1/PPBP is downstream effector of mutant *KRAS* in induction of MPE. These results may suggest CXCL1/PPBP is required for inducing MPE, but it is not clear whether it is sufficient to induce MPE.”**

We appreciate the reviewers' comment. Data presented in Figure 8G clearly suggest that pleural disseminated, mutant *KRAS*/IKK α -addicted tumor cells upregulate and release CXCL1/PPBP in the pleural space that in turn recruit neutrophils in the pleural cavity. However, monocytes are the prevalent immune cell type in the pleural space (Fig 8H) and we have already shown that monocyte chemoattractant CCL2 is an important determinant of tumor cell capacity to induce MPE formation (Stathopoulos GT, JNCI, 2008 an Reference 11 in the present manuscript). Therefore, neutrophilic chemoattractant CXCL1/PPBP is an important mediator of MPE progression but is highly impossible alone to be sufficient for MPE formation. Also, this question was beyond the scope of the present study so we didn't proceed experimentally towards this direction. However, as Reviewer commented it could be worth exploring CXCL1/PPBP overexpression in a *KRAS*^{WT} cell line for their capacity to form MPE. It is worth mentioning that there is no way of presently addressing this experimentally as there is no any commercially available mouse plasmid expressing KC (mouse CXCL1).

8. **“In Figure 9, the survival curve should be shown. Is the inhibitor such as deltarasin very toxic?”**

We appreciate the reviewers' comment. We have now included in Figure 9C the Kaplan-Meier survival plot of this experiment, showing that deltarasin in the concentration used for this experiment was not toxic at all.

9. **“For Figure 10, authors proposed that tumor cells without *KRAS* mutations respond to host cytokines by IKK β -mediated canonical NF- κ B activation resulting in pleural carcinomatosis without malignant pleural effusion (MPE), and mutant *KRAS* cancer cells respond to myeloid IL-1 β in addition to cytokines by IKK α -mediated non-canonical NF- κ B activation leading to in CXCL1/PPBP-effected pleural carcinomatosis with MPE. If so, what is the mechanism? What is the evidence to support such model?”**

We sincerely thank the reviewer for his/her comment. To alleviate the Reviewer's critique, we completely removed Figure 10D.

REVIEWER #2 COMMENTS

“The investigators have previously shown that KRAS mutations are associated with malignant pleural effusions (MPE). In this paper, they have shown that this is dependent on IL-1 signaling that is transduced through IKK α at least in part though increasing CXCL1 gene expression. Their data show that this MPE formation is relatively drug resistant but that concomitant blocking of KRAS and IKK α effectively ameliorates the development of MPE. The work is detailed, comprehensive and using innovative methods. Although some MPE are alleviated by chemotherapy many patients suffer from MPE especially those with limited lung function or excessively large effusions. MPE are notoriously difficult to manage and current management is dependent on palliation and placement of an indwelling tube thoracotomy in patients during their last few weeks of life. The translational application of combination KRAS and IKK α treatment could greatly improve the quality and quantity of the life of patients with terminal cancer with MPE. Although the work is comprehensive, there are some minor concerns that should be addressed.”

We thank the Reviewer for her/his comprehensive evaluation of our manuscript and for recognizing the clinical problem at stake. We have carefully revised the manuscript so as to address all of his/her comments.

- 10. “The introduction indicates that mutant KRAS exerts its pro-MPE effect through CCL2, but the new data show that the effect is exerted through CXCL1 gene expression and is dependent on myeloid cell recruitment. This should be resolved since CCL2 would be expected to recruit monocytes whereas CXCL1 would be expected to recruit neutrophils. The data is ambiguous as to whether monocytes or neutrophils are recruited in the MPE associated with KRAS mutations.”**

We thank the reviewer for this important comment. In our previous publication we incriminated mutant *KRAS* as a promoter of MPE formation and CCL2 secretion that mobilizes myeloid cells from the host bone marrow to the pleural space via the spleen (mainly monocytes). In the present manuscript we continue this work and show that the recruited monocytes in the pleural space secrete a large amount of IL-1 β (Figure 7C). In response to this cytokine pleural disseminated, mutant *KRAS* bearing tumor cells develop an IKK α addiction, resulting in upregulation and release of CXCL1/PPBP chemokines to the pleural space. Consequently, in response to these chemokines neutrophils are starting to accumulate in the pleural space fostering the inflammatory MPE progression. To address the Reviewer’s question as to whether monocytes or neutrophils are recruited in the MPE associated with *KRAS* mutations, we incorporated MPE cytospin data from *C57BL/6*, *Cxcr1*^{-/-}, *Cxcr2*^{+/-} and *Ccr2*^{-/-} mice after tumor inoculation of LLC cells in Figure 8H. The data clearly show that in pleural space during MPE development there is an influx of both monocytes and neutrophils,

with monocytes being the prevalent immune cell type. Additionally, CCL2 recruits monocytes since in *Ccr2*^{-/-} MPEs monocyte population is severely decreased, whereas CXCL1/PPBP recruits neutrophils since in *Cxcr1*^{-/-} and *Cxcr2*^{+/-} MPEs there are much less neutrophils. These issues were clarified throughout the text.

- 11. “In conjunction with the comments above, it would be interesting to know whether monocytes or neutrophils or both are the source of the endogenous IL-1. This is an important point since monocytes produce much more IL-1 per cell than do neutrophils.”**

We thank the reviewer for this comment. We have now included data that clearly show that the main source of IL-1 β secretion in the pleural space during MPE development is monocytes (Figure 7G). Moreover, neutrophils, to a very much lesser extent than monocytes, also secreted IL-1 β cytokine.

- 12. “Although the paper refers to previous references that use the NGL plasmid, I think that this should be better defined, including showing the sequence code of the NF- κ B binding motif, since so much of the data was acquired with this reporter vector.”**

We thank the reviewer for this comment. We have now included all the information, including the sequence code of the NF- κ B binding motif, for the pNGL plasmid, which was extensively used in the present study, in Figure 1A.

- 13. “In addition to showing expression of the reported plasmid as a surrogate marker of the NF- κ B activation pathway, showing binding of RelB/p52 binding to the enhancer region of the IL-1R, by EMSA and ChIP, would strengthen the data set. This would be useful to support the conclusion that KRAS mutation enhances the responsiveness to IL-1 by increasing IL-1R gene expression.”**

We thank the reviewer for this comment. Such an approach would certainly strengthen the microarray and qPCR results of mutant *KRAS* induced increased gene expression of *Il1r1*. However, mutant *KRAS* seems to utilize a different transcription factor than NF- κ B that binds to the enhancement region of *Il1r1* since there are no binding sites for NF- κ B subunits in the promoter region of *Il1r1* gene (EpiTect ChIP qPCR Primers program; Qiagen). For example, AP-1 transcription factor could be an elegant candidate since it has binding sites in the promoter region of *Il1r1* gene and is a known downstream effector of Ras/Raf/Erk signaling pathway. Specifically, in pancreatic adenocarcinoma oncogenic *Kras*^{G12D} induces AP-1 activation that in turns bind to the promoter region of IL-1a, enhancing its expression (Reference 14). Although it became beyond the scope of the present study since NF- κ B is not correlated with the *Il1r1* increased expression, it is a very good experimental approach that will be certainly incorporated in our future studies.

- 14. “The western blots for I κ B α shown in figure 4 is of poor quality and in Figure 5 it appears that immunoreactive I κ B α is depleted by treatment with shChuK and shI κ B α which is not expected and should be explained.”**

We appreciate the reviewers' comment and agree that some western blots in Figure 4K are not of the best quality. However, it is worth mentioning that it was a very demanding Western blot experiment with many different cell protein samples from three different experimental conditions and multiple antibodies tested. We tried our best to successfully complete this and we believe that is of acceptable quality, since all the bands and all differences can be clearly distinguished. Additionally, due to the Reviewer 1 suggestion we incorporated data of EMSA experiments in Figure 4L so as to strengthen the results seen in Figure 4K. I κ B α was indeed depleted when IKK α or IKK β were silenced. Although it seems unexpected it reveals that a long term depletion of an IKK kinase (stable transfection with shRNAs), either IKK α or IKK β , lead the cancer cell to follow compensatory routes to accomplish NF- κ B activation through I κ B α degradation.

- 15. “Similar to the point made above, showing binding of RelB/p52 binding to the enhancer region of the CXCL1, by EMSA and ChIP, would strengthen the data set.”**

We appreciate the reviewers' comment. We incorporated ChIP experiments in Figure 8F that indeed strengthen our microarray and qPCR results, showing that in a *Kras*^{MUT} cell RelB as well as IKK α directly bind to the NF- κ B element in the *Cxcl1* promoter and that IL-1 β reinforces this binding. We used directly ChIP in chromatin from a *Kras*^{MUT} cell line (LLC) treated with PBS or IL-1 β , without EMSA because we wanted to reveal NF- κ B subunit interaction with the *Cxcl1* promoter at intracellular conditions and not on forced conditions, that is high concentration of DNA probe used in EMSA.

- 16. “As pointed out by the investigators, IKK α could serve an epigenetic function through regulating post-translational modifications of histone 3. This could result in enhancement of IL-1R or CXCL1 gene expression and assisted by a histone ChIP. Although this might not be within the scope of this manuscript, the discussion could be elaborated that the combination of the KRAS mutation and epigenetic influences of IKK α in addition to the non-canonical activation of NF- κ B could influence MPE formation.”**

We thank the reviewer for this kind suggestion. We incorporated ChIP assay showing that IKK α directly bind to *Cxcl1* promoter and induce its expression and secretion to the pleural environment. Therefore, epigenetic chromatin modifications of IKK α that result in *Cxcl1* expression is not the case in the present study. However, we cannot exclude that this could be the mechanism used by a *Kras*^{MUT} cancer cell so as to increase *Il1r1* expression and exploit host IL-1 β . We discuss this point in the revised discussion section.

17. “The cartoon shown as Figure 10D is very nice but the labels could be enlarged relative to the drawings.”

We thank the reviewer for this suggestion. However, due to Reviewer 1 comments we completely removed this cartoon from Figure 10.

REVIEWERS' COMMENTS:

Reviewer #1 (Remarks to the Author):

The revised manuscript is much improved. This Reviewer is very appreciative to the efforts of these authors for trying to answer a number of reviewers' questions. However, following issues remain to be discussed.

1. On the basis of the current results of RelB as obtained, does it have any impact on the conclusion?
2. Why the transfection method is used so much? What are the control experiments? What are the pros- and cons of this reporter gene assay based method?
3. It is very disappointing to the comment that authors made in #7 of their point-by point rebuttal. After the mouse entire genome was sequenced and sequence results were released, authors cannot clone a cDNA?

Reviewer #2 (Remarks to the Author):

The author have nicely responded to my original concerns and the manuscript is much improved.

RESPONSE TO REVIEWERS' COMMENTS

REVIEWER #1 (Remarks to the Author):

“The revised manuscript is much improved. This Reviewer is very appreciative to the efforts of these authors for trying to answer a number of reviewers’ questions. However, following issues remain to be discussed.”

We sincerely thank the reviewer for his/her constructive criticism of our work. We have carefully revised the manuscript so as to address all of his/her comments.

- 1. “On the basis of the current results of RelB as obtained, does it have any impact on the conclusion?”**

We thank the reviewer for his/her comment. RelB activation certainly strengthens our conclusion of non-canonical NF- κ B activation in *KRAS* mutant tumors. Also, we showed that RelB nuclear translocation participates in CXCL1 secretion. We have now included RelB results in the Abstract and discussed in more detail in the Discussion section that the non-canonical NF- κ B activation is based on IKK α -RelB activation and that RelB, as IKK α , directly bind to *Cxcl1* promoter contributing to its secretion.

- 2. “Why the transfection method is used so much? What are the control experiments? What are the pros- and cons of this reporter gene assay based method?”**

We appreciate the reviewers’ comment. The present work was mainly elaborated on NF- κ B reporter luciferase gene assay because it constitutes a unique and powerful tool for real-time monitoring NF- κ B activity in cultured cells and in animal models (Blackwell et al 2000; Stathopoulos et al 2006; 2007; 2008; Badr et al 2009; Giannou et al 2015). This assay is highly sensitive and easy to perform with a large number of samples, as in the present study. A great

advantage over all other techniques used for NF- κ B activation is that it is non-invasive and does not require cell harvesting or animal sacrifice, allowing for repetitive measurements over time. The only disadvantage of this assay is that it measures overall NF- κ B activity and cannot provide any information of canonical or non-canonical NF- κ B activation. The control experiment is transfecting cells with a plasmid that constitutively express luciferase gene, such as the pCAG.LUC plasmid that we used in the present study as negative control (Fig. 1b). This control experiment is crucial for determining pathway specific effects and background luciferase activity.

- 3. “It is very disappointing to the comment that authors made in #7 of their point-by point rebuttal. After the mouse entire genome was sequenced and sequence results were released, authors cannot clone a cDNA?”**

We appreciate the reviewers' comment. We believe that our comment was misunderstood. We stated that there is no any commercially available plasmid to immediately perform experiments within the 6-month time limit. Therefore, creating a cDNA clone is feasible but requires time. To alleviate the Reviewer's critique we are pleased to inform him/her that we are presently working upon the role of the specific chemokines in lung cancer and pleural metastasis (Lilis et al; manuscript in preparation) and we commit to incorporate his/her experimental suggestion in the above study so as to elucidate the exact role of CXCL1/PPBP during MPE development.

REVIEWER #2 (Remarks to the Author):

“The author have nicely responded to my original concerns and the manuscript is much improved.”

We deeply appreciate the efforts spent by the Reviewer in order to improve our manuscript.